# Rethinking Cross-Domain Sequential Recommendation under Open-World Assumptions

Submission Id: 242*

## ABSTRACT

Cross-Domain Sequential Recommendation (CDSR) methods aim to tackle the data sparsity and cold-start problems present in Single-Domain Sequential Recommendation (SDSR). Existing CDSR works design their elaborate structures relying on overlapping users to propagate the cross-domain information. However, current CDSR methods make closed-world assumptions, assuming fully overlapping users across multiple domains and that the data distribution remains unchanged from the training environment to the test environment. As a result, these methods typically result in lower performance on online real-world platforms due to the data distribution shifts. To address these challenges under open-world assumptions, we design an **A**daptive **M**ulti-**I**nterest **D**ebiasing framework for cross-domain sequential recommendation (**AMID**), which consists of a multi-interest information module (**MIM**) and a doubly robust estimator (**DRE**). Our framework is adaptive for open-world environments and can improve the model of most off-the-shelf single-domain sequential backbone models for CDSR. Our MIM establishes interest groups that consider both overlapping and non-overlapping users, allowing us to effectively explore user intent and explicit interest. To alleviate biases across multiple domains, we developed the DRE for the CDSR methods. We also provide a theoretical analysis that demonstrates the superiority of our proposed estimator in terms of bias and tail bound, compared to the IPS estimator used in previous work. To promote related research in the community under open-world assumptions, we collected an industry financial CDSR dataset from Alipay, called **"MYbank-CDR"**. Extensive offline experiments on four industry CDSR scenarios including the Amazon and MYbank-CDR datasets demonstrate the remarkable performance of our proposed approach. Additionally, we conducted a standard A/B test on Alipay, a large-scale financial platform with over one billion users, to validate the effectiveness of our model under open-world assumptions.

## CCS CONCEPTS

• **Information systems** → **Recommender systems**; • **Computing methodologies** → **Neural networks**.

## KEYWORDS

Open-world Assumptions, Cross-Domain Sequential Recommendation, Sequential Recommendation

## 1 INTRODUCTION

Single-domain sequential recommendation (SDSR) has garnered wide attention in E-commerce recommender systems, for its ability to model dynamic user preferences based on their sequential interactions. Nevertheless, traditional sequential recommendation methods are often limited by data sparsity and the cold-start problem, affecting their recommendation performance [12, 17, 38].

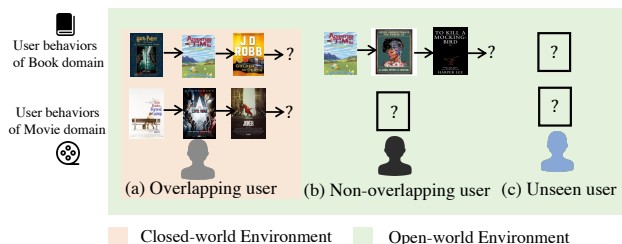

**Figure 1: While traditional methods [27, 49, 50] focus only on overlapping users (a) and a few methods [2, 19, 20] can handle non-overlapping users (b), they still have some limitations. However, our method not only considers users (a) and (b), but also assigns importance to unseen users (c).**

To alleviate these problems, several Cross Domain Sequential Recommendation (CDSR) approaches [14, 27, 29] have been proposed to transfer the knowledge between the source domain and target domain. Existing CDSR approaches commonly employ feature combination [27, 49, 52] or bi-directional transfer mapping strategies [19, 20, 50] to enhance recommendation accuracy by leveraging the overlapping users. Most of these methods conduct experiments under closed-world assumptions, where multiple domains share fully overlapping users and data distribution remains unchanged from the training environment to the test environment. However, such closed-world assumptions do not exist in industry recommendation platforms, which operate under open-world assumptions instead. In this study, the concept of the "open-world is used in contrast to the closed world. The term "open-world" [44, 48] is employed to describe and summarize the scenario, which contains minority overlapping users among multiple domains and contains the selection bias in modeling. In the open-world environment, the input/output space expands, and data distribution shifts due to unseen factors. Therefore, to enhance the model's performance in CDSR, it is imperative to address the challenges that arise under open-world assumptions.

First of all, *for multiple domains with few overlapping users, how to construct the model with the non-overlapping users and improve the recommendation performance?* Most previous CDSR methods cannot be directly extended to handle partially overlapping CDSR settings, especially when there are only a few shared users across domains that are often encountered in the real world. To address this challenge, recent researches [6, 9, 20, 26] have explored the use of graphic deep learning to enhance both overlapping and non-overlapping user embeddings and propagate interest information among non-overlapping users. More recently, Cao et al. [2] have designed a graph neural network with a contrastive cross-domain infomax objective to improve the learning of non-overlapping user embeddings. Nevertheless, in partially overlapping CDSR scenarios,

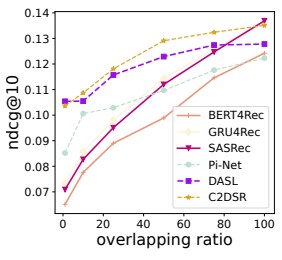
(a) Music Domain

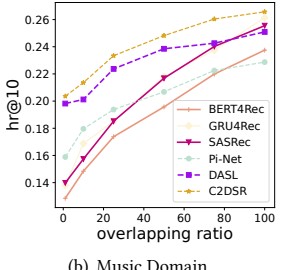
(b) Music Domain

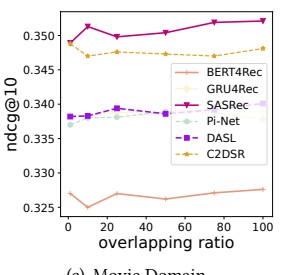
(c) Movie Domain

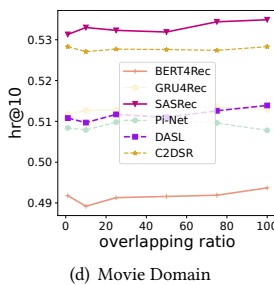
(d) Movie Domain

Figure 2: Solid lines denote the SDSR methods, while dashed lines denote the CDSR methods. Due to the lack of abundant overlapping users, SASRec (SDSR) outperforms all the CDSR methods in the Movie domain.

such methods have great limitations and will drop their performance under open-world assumptions. These CDSR approaches still rely heavily on overlapping users, with more than 70% of users being common across domains, to establish bridge connections among multiple domains and perform knowledge aggregation and transition processes.

The second challenge arises from an observation in open-world scenarios: Recommender systems frequently engage primarily with active users. In the training phase, some unexposed users who are not exposed to the platform, and thus unbeknownst to the models, may surface in the testing stage. This issue causes a performance drop after adapting from an offline to an online environment [30, 31, 46]. So, the second challenge is *how to alleviate the selection bias of the model in the environment with the data distribution shift?* CaseQ [46] learns context-specific representations of sequences to capture temporal patterns in various environments. Similarly, DCRec [47] proposes a novel debiasing contrastive learning paradigm to address the popularity bias issue in single-domain recommendation systems. However, these methods overlook the selection bias among the domains that exists in open-world scenarios, leading to biased performance estimation. To address this selection bias, [21] proposes an Inverse-Propensity-Score (IPS) estimator, yet often suffering high variance.

In this paper, we first rethink cross-domain sequential recommendation under open-world assumptions and identify the primary challenges. To address these challenges, we propose an adaptive multi-interest debiasing framework for CDSR, which includes a multi-interest information module and a doubly robust estimator. **Our contributions are as follows.**

**1)** To our best knowledge, this paper is the first effort addressing the open-world challenges in CSDR. We conduct empirical analysis and show that extending existing CDSR models to the open-world environment yields two primary challenges that used to be overlooked: i) How to construct a model in scenarios where the majority of users are non-overlapping, without relying on overlapping users? and ii) How to eliminate selection bias of the model with the data distribution shift?

**2)** We design an **A**daptive **M**ulti-**I**nterest **D**ebiasing framework for cross-domain sequential recommendation (**AMID**), which could be integrated with most off-the-shelf SDSR methods [12, 17, 38]. It is composed of a multi-interest information module (**MIM**) and

a doubly robust estimator (**DRE**). MIM transfers cross-domain information for both overlapping and non-overlapping users, while DRE eliminates selection bias and popularity bias to obtain unbiased performance estimation. Moreover, we provide a theoretical analysis that demonstrates the superiority of DRE in terms of bias and tail bound, compared to the IPS estimator used in [21].

**3)** In order to foster further research in the community, particularly under open-world assumptions, we gathered a real-world financial CDSR dataset from Alipay, called **"MYbank-CDR"**. As far as we are aware, "MYbank-CDR" is the first publicly available cross-domain financial dataset. The collected dataset "MYbank-CDR" and the source code will be made publicly available upon acceptance.

**4)** We demonstrate that our proposed **AMID**, when integrated with multiple single-domain sequential recommendation models, achieves state-of-the-art results compared to CDR, CDSR and debiasing methods. Additionally, we conduct online experiments to validate the performance of our proposed framework in a real-world CDSR financial platform with millions of daily traffic logs.

## 2 MOTIVATION: TOWARDS OPEN-WORLD CDSR

Current CDSR methods conduct their experiments under closed-world assumptions which assume that there exist fully or mostly overlapping users across domains. However, in real-world applications, the number of overlapping users across domains is typically a minority. To validate their performance in the open-world environment, we perform motivational experiments on the Amazon dataset with three single-domain sequential recommendation methods (BERT4Rec [38], GRU4Rec [12] and SASRec [17]) and three cross-domain sequential recommendation methods (Pi-Net [27], DASL [19] and C$^2$DSR [2]). Following previous works [24, 45], we vary the overlapping ratio [1] to simulate different CDSR scenarios. We present a plot of the performance of the models from two domains in Fig 2. In the Movie domain, SASRec (SDSR) achieves the best performance. Similarly, in the Music domain with a 100% overlapping ratio, SASRec (SDSR) outperforms DASL (CDSR). This is attributed to the fact that existing CDSR methods rely on overlapping users to construct their models or transfer information across domains, which leads to a decrease in performance in a partially

---

[1]It should be noticed that the overlapping ratio is not equivalent to the proportion of overlapping users, but rather controls the number of existing overlapping users of the training set.

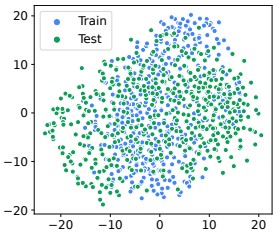

(a) Ratio: 25%, Music

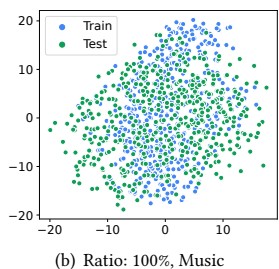

(b) Ratio: 100%, Music

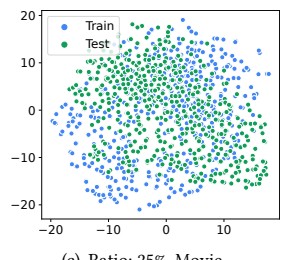

(c) Ratio: 25%, Movie

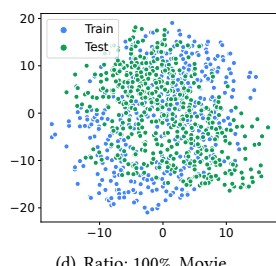

(d) Ratio: 100%, Movie

**Figure 3: Selection bias can cause a distribution shift in the cross-domain sequential scenario with few overlapping users (control ratio is 25%).**

overlapping scenario. Hence, this finding serves as a motivating factor for us to design a high-performing CDSR model that can be applied in the open-world environment (**1st challenge**). Furthermore, a counterintuitive phenomenon has been observed, where SDSR methods exhibit a lesser reliance on overlapping users compared to CDSR methods. In an ideal scenario, SDSR methods should exhibit inferior performance compared to CDSR methods when the overlapping user ratio is high. However, experimental results have revealed that the performance gap actually diminishes as the ratio decreases. Surprisingly, when the ratio is small, the performance of a few SDSR methods outperforms CDSR methods, and in some cases, even when the ratio is high.

To analyze the underlying cause for this phenomenon, we try to visualize the distribution shift between the train set and the test set. Figure 6 displays the t-SNE visualization of user embeddings [2] for the train set (blue dots) and the test set (green dots). Training with a 25% proportion simulates the situation where the training data in a real-world scenario may suffer from unseen factors. We observed that the distribution difference at 25% proportion is larger than that at 100% proportion. These visualization results confirm that the distribution shift from training to testing indeed exists in the CDSR scenarios. The distribution shift often occurs in a real-world platform under open-world assumptions. In real-world recommendation systems, users to be exposed are sometimes selected by the recommendation algorithm based on factors such as estimated conversion rates and business rules. During training, only the data of these exposed users are used because their interaction labels are considered meaningful, while the unexposed users are overlooked. However, during the inference stage, estimated conversion scores are required for all users, including the unexposed ones, in order to determine the selection of users to be exposed in the recommendation system. The rating data of these unexposed users are missing not at random [4], leading to selection bias in the CDSR scenario. Therefore, this raises another question: "How can we mitigate data selection bias across multiple domains in the open-world environment?" (**2nd challenge**)

## 3 PRELIMINARIES

### 3.1 Problem Definition

In this paper, we consider a partially overlapping CDSR scenario composed of multiple domains $\mathcal{Z} = \{Z_1, ..., Z_{|\mathcal{Z}|}\}$. Let $\mathcal{U} = \{u_1, ...,$

---

[2]The trained DASL [19] model is used to generate user embeddings.

$u_{|\mathcal{U}|}\}$, $\mathcal{V} = \{v_1^{Z_1}, ..., v_{|\mathcal{V}|}^{Z_{|\mathcal{Z}|}}\}$ be the user set, item set, rating set. A user who only has historical behaviors in one domain is referred to as a non-overlapping user, while a user with historical behaviors in multiple domains is referred to as an overlapping user. As a certain user, denote $\mathcal{S} = \{S^{Z_1}, ..., S^{Z_{|\mathcal{Z}|}}\}$ the corresponding sequential behaviours of the users. For example, $S^{Z_1} = \{v_1, ..., v_T\}$ represents the single-domain sequence, where $T$ is a variable length. Given the data $\mathcal{D} = \mathcal{U} \times \mathcal{V}$, CDSR aims to develop a personalized ranking function that utilizes the past item sequences from multiple domains of a user and predicts the next item (i.e. $v_{T+1}$) in each domain that the user is most likely to choose.

Different from conventional cross-domain recommendation (CDR), CDSR methods pay more attention to modeling sequential behavior dependencies. Mathematically, the objective of CDR methods are formulated as follows:

$$\operatorname*{argmax}\ P^X\left(r_{u,v}^X = v | U^X, U^Y, V^X\right), \text{if}\ v \in \mathcal{V}^X. \quad (1)$$

where $r_{u,v}$ denotes the prediction from user $u$ to item $v$ in domain $X$. However, the objective of CDSR approaches is to predict the next item for a given user $u$ based on their interaction sequences:

$$\operatorname*{argmax}\ P^X\left(v_{|S^X|+1}^X = v | S^X, S^Y, U^X, U^Y, V^X\right), \text{if}\ v \in \mathcal{V}^X. \quad (2)$$

### 3.2 Causal Graph

To tackle the issue of incomplete and insufficient observed information, we construct a causal view and propose an adaptive multi-interest debiasing framework. $\mathcal{S}_U^Z$, $\mathcal{R}_U^Z$, $SC^Z$, and $GC$ denote the random variables of the historical event sequence, the observed ratings, the domain-specific confounder, and the general confounder. $O$, which represents the observed variable of the instance ($O = 1$, observed; $O = 0$, unobserved), is decided by $O^{Z_1}$ and $O^{Z_2}$ commonly. If a user is not observed in either of the two domains, $O$ is 0; otherwise, it is 1. The SDSR methods focus on learning the specified confounder $SC^Z$ for a given user, while the previous CDSR approaches construct models based on overlapping users to obtain the general confounder $GC$. The links between $(SC^Z, GC) \rightarrow \mathcal{R}_U^Z$ represent the causal effect of the domain-specific and general confounders on their interaction label. In observational studies [11, 28, 37], the collected rating data is often unevenly presented, and the variables $\mathcal{S}_U^Z$ and $\mathcal{R}_U^Z$ can affect the observation $O$ of a given instance. This mechanism gives rise to selection bias, resulting in an inconsistent

distribution of the observed rating data as compared to the ideal test distribution.

From another perspective, the causal graph depicts two sources of association between the causes $S$ and the outcome $R$: (1) the desirable causal effect $S \rightarrow (SC, GC) \rightarrow R$; (2) the collision path $S \rightarrow O \leftarrow R$ that connects $S$ and $R$ through their common (conditioned on) effects $O = 1$. Analyses conditioned on $O = 1$ may generate spurious associations between $S$ and $R$. It is important to note that the domains $Z_1$ and $Z_2$ commonly influence the observed variable. Models learned from observed data may be affected by the issue of selection bias. In the open-world environment, this is a critical issue that must be addressed to prevent a drop in online performance.

### 3.3 Single-domain Sequential Recommendation Methods

In this research, our primary objective is to develop a universal cross-domain structure that can improve the performance of single-domain sequential recommendation models by seamlessly integrating them into comprehensive CDSR models. To achieve this, we first explore the fundamental mechanisms of the single-domain sequential recommendation network.

**Embedding layer.** We map the fixed-length sequence $S = \{v_1, ..., v_T\}$ to a $d$-dimensional embedding space, which is achieved through truncation or padding. Besides, a learnable parameter position embedding matrix is utilized to enhance the chronologically ordered information of the sequence. Specially, we get the sequence embedding $\mathbf{S}_u = \{\mathbf{h}'_{v_1}, ..., \mathbf{h}'_{v_T}\} \in \mathbb{R}^{T \times d}$.

**Sequential information encoder.** In the previous single-domain sequential recommendation methods, they designed various sequential information encoders to extract short-/long-term item relationships among the sequence. For example, GRU4Rec [12] designs multiple GRU layers for the sequential recommendation. SASRec [17] proposes stacking self-attention blocks with residual connections, while BERT4Rec [38] develops the deep bidirectional self-attention to model user behavior sequences. For convenience, we obtain the enhanced sequential embedding via a function $\mathcal{F}$ which represents their designed sequential information encoder. This process is formulated as $\mathbf{h}_{v_1}, ..., \mathbf{h}_{v_T} = \mathcal{F}(\mathbf{h}'_{v_1}, ..., \mathbf{h}'_{v_T})$.

## 4 METHODOLOGY

### 4.1 Multi-interest Information Module

In this section, we will introduce our multi-interest information module, which can convert a normal SDSR model into a CDSR model. The module consists of two steps: interest group construction and information propagation. For simplicity, we only include two domains as examples and show the operation on them, but our methods can be applied to multiple domains. Our design is motivated by the goal of creating interest groups by identifying users with similar preferences and sharing information as widely as possible.

**Group construction.** Initially, we compute the group flag by evaluating the similarity between their sequences point-by-point. For instance, consider two sequences $S_{u_i}^{Z_1} \in \mathbb{R}^{T \times d}$ and $S_{u_j}^{Z_2} \in \mathbb{R}^{T \times d}$ from users $u_i$ and $u_j$ in different domains. We then determine the

group flag among users as follows:

$$\mathbf{a}'_{ij} = \max[(S_{u_i}^{Z_1} \mathbf{W}_1)(S_{u_j}^{Z_2} \mathbf{W}_2)^\top] \tag{3}$$

where $\mathbf{W}_1, \mathbf{W}_2 \in \mathbb{R}^{d \times d}$ are the transformation matrix. The max function is utilized to find the nearest similarity between user $u_i$ and $u_j$ from $T \times T$ similarity relations. Last, we get the group flag via the threshold determination where $\mathbf{a}_{ij} = 1$ means the users $u_i$ and $u_j$ are in the same group.

$$\mathbf{a}_{ij} = \begin{cases} 0, & \mathbf{a}'_{ij} < k \\ 1, & \mathbf{a}'_{ij} \geq k \end{cases} \tag{4}$$

**Information propagation.** After constructing the interest groups, the cross-domain information will propagate among the same group. The cross-domain message $\mathbf{m}_{u_i^{Z_1} \leftarrow u_j^{Z_2}} \in \mathbb{R}^{T \times d}$ can be obtained:

$$\mathbf{m}_{u_i^{Z_1} \leftarrow u_j^{Z_2}} = \mathbf{a}_{ij} \cdot (S_{u_j}^{Z_2} \mathbf{W}_{ip}) \tag{5}$$

where $\mathbf{W}_{ip} \in \mathbb{R}^{d \times d}$ is the trainable parameter to transfer the cross-domain knowledge. When the user has the behaviors in multiple domains, the message from the same user in a different domain (e.g. $\mathbf{m}_{u_i^{Z_1} \leftarrow u_i^{Z_2}}$) will also be propagated. For the target user (e.g. $u_i^{Z_1}$), we concatenate all the information from other domains along the last dimension to obtain the aggregated message $\mathbf{m}'_{u_i^{Z_1}} \in \mathbb{R}^{T \times d \times N}$, where $N$ is the number of the sampled users. We then use transformation matrix $\mathbf{W}_C \in \mathbb{R}^{N \times 1}$ and $\mathbf{W}_F \in \mathbb{R}^{d \times d}$ to fuse the information and get the enhanced sequence representation $S_{u_i}^{*Z} \in \mathbb{R}^{2T \times d}$ concatenated with the initial sequence information, where the Squeeze function reduces the dimension of the matrix. **For users in other domains, information propagation occurs in a similar manner.**

$$S_{u_i}^{*Z} = \text{Concat}(S_{u_i}^Z, \text{Squeeze}(\mathbf{m}'_{u_i^Z} \mathbf{W}_C))\mathbf{W}_F \tag{6}$$

### 4.2 Prediction Layer

We construct a prediction layer to estimate the user's $u_i$ preference towards the target item $v_k$ as:

$$\hat{r}_{u_i, v_k}^Z = \sigma(\text{MLPs}(\text{Mean}(S_{u_i}^{*Z}) || \mathbf{v}_k^Z)) \tag{7}$$

where MLPs consist of stacked MLP layers that take as input the concatenation of enhanced sequence embedding and item embedding. The sigmoid function is denoted by $\sigma$, and the Mean function averages the embedding along the temporal dimension.

### 4.3 Doubly Robust Estimator for CDSR

In this part, we propose a novel doubly robust estimator, which generalizes the traditional DR estimator [42] to the cross-domain sequential scenarios. Suppose $\hat{\mathbf{R}}^Z \in \mathbb{R}^{\mathcal{U}^Z \times \mathcal{V}^Z}$ be a prediction matrix and $\mathbf{R}^Z \in \mathbb{R}^{\mathcal{U}^Z \times \mathcal{V}^Z}$ be a true rating matrix, the prediction inaccuracy $\mathcal{P}$ and the doubly robust estimator $\mathcal{E}_{\text{DR}}^*$ are defined as:

$$\mathcal{P} = \frac{1}{|\mathcal{Z}|} \sum_{Z \in \mathcal{Z}} \frac{1}{|\mathcal{D}^Z|} \sum_{u,v \in \mathcal{D}^Z} e_{u,v}^Z. \tag{8}$$

$$\mathcal{E}_{\text{DR}}^* = \frac{1}{|\mathcal{Z}|} \sum_{Z \in \mathcal{Z}} \frac{1}{|\mathcal{D}^Z|} \sum_{u,v \in \mathcal{D}^Z} \left( \hat{e}_{u,v}^Z + \frac{o_{u,v}^Z \delta_{u,v}^Z}{\hat{p}_{u,v}^Z} \right). \tag{9}$$

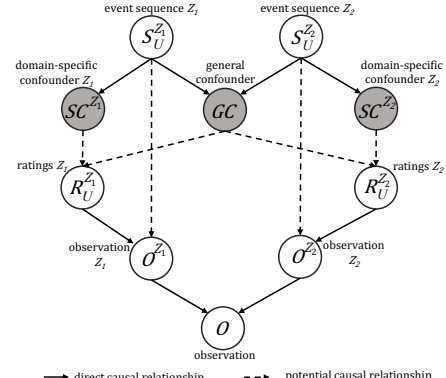

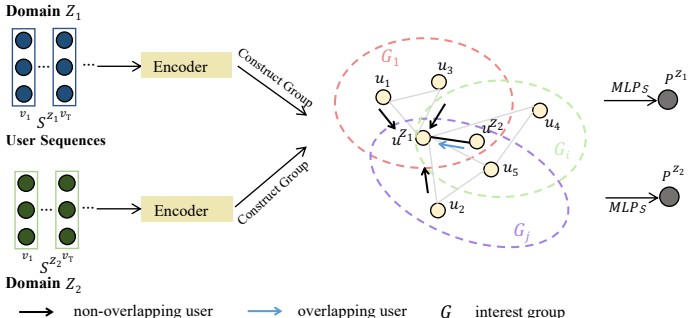

**Figure 4: The casual graph of the selection bias in CDSR. Grey and white variables represent latent and observed variables, respectively.**

**Figure 5: Overview of our multi-interest information module. The encoder denotes the sequential information encoder from the SDSR model. The black (i.e., $u^{Z_1} \rightarrow u_1$) and blue (i.e., $u^{Z_1} \rightarrow u^{Z_2}$) solid arrow denote two different types of messages propagated by the different users and the same user in different domains.**

where $e_{u,v}^Z = |\hat{r}_{u,v}^Z - r_{u,v}^Z|$ or $e_{u,v}^Z = (\hat{r}_{u,v}^Z - r_{u,v}^Z)^2$ via optional measure metrics for MAE or MSE. The imputation error $\hat{e}_{u,v}^Z = g_{\phi^Z}(\text{Mean}(S_u^{*Z})||\mathbf{v}^Z))$ is computed by imputation model which aims to estimate the prediction error $e_{u,v}$ on the observed data. We also learn the propensity $\hat{p}_{u,v} = g_{\psi^Z}(\text{Mean}(S_u^{*Z})||\mathbf{v}^Z))$. The imputation model $g_{\phi^Z}$ and the propensity model $g_{\psi^Z}$ are implemented in a multi-task manner. The bias of the estimator is derived as follows.

**Lemma 4.1 (Bias of DR Estimator)**. Given imputation errors $\hat{\mathbf{E}}^Z$ and learned propensities $\hat{\mathbf{P}}^Z$ for all user-item pairs, the bias of the DR estimator in the CDSR task is

$$\text{Bias}(\mathcal{E}_{DR}^*) = \frac{1}{|\mathcal{Z}|} \sum_{Z \in \mathcal{Z}} \left[ \frac{1}{|\mathcal{D}^Z|} \left| \sum_{u,v \in \mathcal{D}^Z} \Delta_{u,v}^Z \delta_{u,v}^Z \right| \right] \quad (10)$$

where the imputation error $\delta_{u,v}^Z$ and the learned propensities $\Delta_{u,v}^Z$ is defined as:

$$\Delta_{u,v}^Z = \frac{\hat{p}_{u,v}^Z - p_{u,v}^Z}{\hat{p}_{u,v}^Z}, \quad \delta_{u,v}^Z = e_{u,v}^Z - \hat{e}_{u,v}^Z \quad (11)$$

**Corollary 4.1 (Double Robustness)**. The DR estimator for CDSR is unbiased when either imputed errors $\hat{\mathbf{E}}^Z$ or learned propensities $\hat{\mathbf{P}}^Z$ are accurate for all user-item pairs.

**Lemma 4.2 (Tail Bound of DR Estimator)**. Given imputation errors $\hat{\mathbf{E}}^Z$ and learned propensities $\hat{\mathbf{P}}^Z$, for any prediction matrix $\hat{\mathbf{R}}^Z$, with probability 1-$\eta$, the deviation of the DR estimator from its expectation has the following tail bound in CDSR task.

$$\left| \mathcal{E}_{DR}^* - \mathbb{E}_O[\mathcal{E}_{DR}^*] \right| \leq \sqrt{\frac{\log(\frac{2}{\eta})}{2|\mathcal{Z}|(\sum_{Z \in \mathcal{Z}} |\mathcal{D}^Z|)^2} \sum_{Z \in \mathcal{Z}} \left[ \frac{1}{|\mathcal{D}^Z|} \sum_{u,v \in \mathcal{D}^Z} \left( \frac{\delta_{u,v}^Z}{\hat{p}_{u,v}^Z} \right)^2 \right]} \quad (12)$$

**Corollary 4.2 (Tail Bound Comparison)**. Suppose imputed errors $\hat{\mathbf{E}}^Z$ are such that $0 \leq \hat{e}_{u,v}^Z \leq 2e_{u,v}^Z$ for each $u,v \in \mathcal{D}^Z$, then for any learned propensities $\hat{\mathbf{P}}$, the tail bound of the proposed estimator will be lower than that of the IPS estimator which is utilized in IPSCDR [21]. The proof of the lemmas and the corollaries are demonstrated in Appendix A.

## 4.4 Joint learning

We apply an alternating training for joint learning. In the first step, we train the imputation and prediction models on the observed data by minimizing the proposed hybrid loss in the first step.

$$\mathcal{L}_e(\theta, \phi, \psi) = \frac{1}{|\mathcal{Z}|} \sum_{Z \in \mathcal{Z}} \left( \frac{1}{|O^Z|} \sum_{u,v \in O^Z} e_{u,v} + \lambda_1 \sum_{u,v \in O^Z} \frac{(\hat{e}_{u,v} - e_{u,v})^2}{\hat{p}_{u,v}} \right) \quad (13)$$

$$+ \lambda_2 ||\theta||_F^2 + \lambda_3 ||\phi||_F^2 + \lambda_4 ||\psi||_F^2 \quad (14)$$

The Frobenius norm $||.||_F^2$ is used to measure the norm of matrices. The hyperparameters $\lambda_{1,2,3,4}$ are used to control the trade-off between regularization and the multiple loss. After training the model on the observed data, we continue to train our prediction model $\theta$ on $\mathcal{D}^Z$ to alleviate the bias.

$$\mathcal{L}_r(\theta, \phi, \psi) = \frac{1}{|\mathcal{Z}|} \sum_{z \in \mathcal{Z}} \left[ \frac{1}{|\mathcal{D}^z|} \sum_{u,v \in \mathcal{D}^z} \left( \hat{e}_{u,v} + \frac{o_{u,v}(e_{u,v} - \hat{e}_{u,v})}{\hat{p}_{u,v}} \right) \right] \quad (15)$$

$$+ \lambda_5 ||\theta||_F^2 \quad (16)$$

The label $y_{u,v}$ is unavailable for the data point in the set $\mathcal{D}^Z - O^Z$. $\lambda_5$ balances the regularization term. The learning process of our framework is shown in Appendix B.

## 5 EXPERIMENTS

### 5.1 Experimental Setup

To conduct our experiments, we followed the methodology of previous research [2, 45] and used the Amazon datasets [3], which consist of 24 item domains. We selected two pairs of domains, namely "Cloth-Sport" and "Phone-Elec", and formulated two different tasks. Besides, we collect a Table 3 summarizes the statistics for each task. To replicate selection bias in online platforms, we adjusted the non-overlapping ratio $\mathcal{K}_u$ for each dataset, selecting from {25%, 75%}. Varying the ratio results in different numbers of non-overlapping

---

[3]http://jmcauley.ucsd.edu/data/amazon/index 2014.html

**Table 1: Experimental results (%) on the bi-directional Cloth-Sport and Phone-Elec CDSR scenario with different $\mathcal{K}_u$.**

| Methods | Cloth-domain recommendation | | | | Sport-domain recommendation | | | | Phone-domain recommendation | | | | Elec-domain recommendation | | | |
|---|---|---|---|---|---|---|---|---|---|---|---|---|---|---|---|---|
| | $\mathcal{K}_u$=25% | | $\mathcal{K}_u$=75% | | $\mathcal{K}_u$=25% | | $\mathcal{K}_u$=75% | | $\mathcal{K}_u$=25% | | $\mathcal{K}_u$=75% | | $\mathcal{K}_u$=25% | | $\mathcal{K}_u$=75% | |
| | NDCG@10 | HR@10 | NDCG@10 | HR@10 | NDCG@10 | HR@10 | NDCG@10 | HR@10 | NDCG@10 | HR@10 | NDCG@10 | HR@10 | NDCG@10 | HR@10 | NDCG@10 | HR@10 |
| BERT4Rec [38] | 1.42±0.13 | 2.96±0.15 | 2.28±0.19 | 4.43±0.46 | 2.96±0.25 | 5.60±0.34 | 4.19±0.18 | 7.77±0.24 | 6.13±0.16 | 10.87±0.24 | 6.20±0.17 | 11.07±0.47 | 8.24±0.22 | 13.05±0.33 | 10.58±0.07 | 17.28±0.05 |
| GRU4Rec [12] | 2.03±0.23 | 4.19±0.35 | 3.23±0.15 | 6.18±0.37 | 3.63±0.22 | 7.35±0.47 | 4.85±0.16 | 9.43±0.20 | 6.99±0.23 | 12.98±0.39 | 6.98±0.18 | 12.90±0.38 | 10.11±0.12 | 16.77±0.23 | 11.41±0.08 | 19.00±0.12 |
| SASRec [17] | 2.00±0.13 | 4.26±0.23 | 3.34±0.18 | 6.66±0.38 | 3.69±0.32 | 7.46±0.55 | 4.96±0.30 | 9.69±0.38 | 7.19±0.19 | 13.32±0.30 | 7.14±0.26 | 13.34±0.34 | 10.56±0.18 | 17.77±0.21 | 11.64±0.04 | 19.67±0.18 |
| STAR [35] | 1.97±0.27 | 4.19±0.39 | 3.37±0.15 | 6.54±0.28 | 3.40±0.29 | 7.10±0.51 | 4.77±0.18 | 9.29±0.15 | 6.96±0.24 | 13.30±0.45 | 7.13±0.18 | 13.71±0.51 | 9.72±0.17 | 16.38±0.26 | 11.18±0.13 | 19.00±0.29 |
| MAMDR [25] | 2.11±0.09 | 4.25±0.17 | 3.44±0.15 | 6.60±0.17 | 3.53±0.44 | 7.21±0.24 | 4.84±0.14 | 9.33±0.42 | 7.01±0.22 | 13.38±0.42 | 7.19±0.19 | 13.81±0.20 | 9.79±0.31 | 16.47±0.39 | 11.24±0.47 | 19.08±0.25 |
| SSCDR [16] | 2.02±0.18 | 4.21±0.36 | 3.42±0.24 | 6.57±0.17 | 3.45±0.24 | 7.14±0.40 | 4.81±0.29 | 9.31±0.57 | 6.99±0.39 | 13.35±0.45 | 7.21±0.35 | 13.83±0.30 | 9.75±0.27 | 16.45±0.27 | 11.27±0.31 | 19.13±0.33 |
| Pi-Net [27] | 1.84±0.26 | 3.82±0.38 | 2.77±0.20 | 5.56±0.32 | 3.37±0.17 | 7.05±0.22 | 4.56±0.25 | 8.81±0.34 | 7.02±0.26 | 12.68±0.50 | 7.11±0.26 | 12.95±0.36 | 10.00±0.11 | 16.33±0.28 | 11.64±0.06 | 19.34±0.26 |
| DASL [19] | 2.11±0.29 | 4.56±0.43 | 3.32±0.17 | 6.62±0.29 | 3.96±0.16 | 8.11±0.37 | 4.81±0.39 | 9.71±0.51 | 7.10±0.15 | 13.29±0.17 | 7.22±0.13 | 13.22±0.17 | 10.35±0.33 | 17.15±0.57 | 11.72±0.07 | 19.52±0.18 |
| C$^2$DSR [2] | 2.27±0.18 | 4.73±0.35 | 3.31±0.07 | 6.62±0.24 | 3.74±0.26 | 7.99±0.42 | 5.18±0.10 | 10.31±0.07 | 7.54±0.15 | 14.04±0.23 | 7.30±0.19 | 13.96±0.44 | 10.71±0.13 | 17.98±0.10 | 11.74±0.03 | 20.04±0.21 |
| DCRec [47] | 2.22±0.14 | 4.37±0.12 | 3.49±0.08 | 6.51±0.12 | 3.85±0.14 | 7.72±0.27 | 5.21±0.15 | 10.08±0.26 | 7.18±0.08 | 13.58±0.13 | 7.15±0.07 | 13.21±0.25 | 10.48±0.14 | 17.84±0.22 | 11.79±0.08 | 20.01±0.09 |
| BERT4Rec [38] + CaseQ [46] | 2.24±0.16 | 4.41±0.13 | 3.45±0.07 | 6.48±0.13 | 3.82±0.15 | 7.69±0.29 | 5.28±0.16 | 10.04±0.27 | 7.25±0.07 | 13.52±0.14 | 7.19±0.06 | 13.33±0.26 | 10.56±0.13 | 17.77±0.23 | 11.88±0.07 | 19.98±0.10 |
| GRU4Rec [12] + CaseQ [46] | 2.20±0.15 | 4.50±0.25 | 3.58±0.09 | 6.57±0.19 | 3.98±0.18 | 7.76±0.46 | 5.30±0.14 | 10.08±0.18 | 7.28±0.18 | 13.34±0.29 | 7.38±0.17 | 13.52±0.28 | 10.59±0.15 | 17.88±0.23 | 11.98±0.07 | 20.04±0.14 |
| SASRec [17] + CaseQ [46] | 2.23±0.23 | 4.46±0.12 | 3.51±0.07 | 6.56±0.26 | 3.87±0.14 | 7.75±0.46 | 5.32±0.16 | 10.21±0.36 | 7.31±0.12 | 13.47±0.23 | 7.37±0.11 | 13.49±0.15 | 10.60±0.12 | 17.77±0.16 | 11.93±0.06 | 19.98±0.14 |
| BERT4Rec [38] + IPSCDR [21] | 1.95±0.11 | 3.96±0.20 | 2.82±0.31 | 5.55±0.73 | 3.65±0.22 | 7.06±0.33 | 4.93±0.29 | 9.46±0.21 | 7.29±0.19 | 13.20±0.32 | 7.47±0.33 | 13.46±0.67 | 9.71±0.24 | 15.75±0.30 | 11.59±0.15 | 19.66±0.20 |
| GRU4Rec [12] + IPSCDR [21] | 2.53±0.13 | 4.93±0.44 | 3.79±0.18 | 7.11±0.42 | 4.24±0.28 | 8.50±0.36 | 5.64±0.16 | 10.60±0.20 | 7.58±0.22 | 13.90±0.49 | 7.84±0.26 | 14.16±0.54 | 10.69±0.07 | 17.78±0.12 | 12.18±0.10 | 20.33±0.17 |
| SASRec [17] + IPSCDR [21] | 2.48±0.10 | 4.92±0.22 | 3.67±0.08 | 6.82±0.22 | 4.10±0.26 | 8.13±0.46 | 5.55±0.15 | 10.59±0.26 | 7.61±0.15 | 14.05±0.15 | 7.95±0.14 | 14.27±0.19 | 10.80±0.12 | 18.03±0.21 | 12.26±0.12 | 20.36±0.32 |
| BERT4Rec [38] + MIM | 1.80±0.23 | 3.68±0.41 | 2.24±0.13 | 4.48±0.20 | 2.83±0.18 | 5.45±0.33 | 4.11±0.31 | 7.79±0.47 | 6.83±0.13 | 11.81±0.25 | 6.84±0.14 | 12.11±0.34 | 8.41±0.13 | 13.38±0.16 | 11.23±0.17 | 18.39±0.34 |
| GRU4Rec [12] + MIM | 2.58±0.07 | 4.88±0.09 | 3.63±0.13 | 6.61±0.30 | 4.47±0.30 | 8.50±0.40 | 5.62±0.18 | 10.58±0.25 | 7.88±0.23 | 14.31±0.34 | 7.75±0.32 | 14.39±0.30 | 11.06±0.17 | 18.46±0.40 | 12.30±0.17 | 20.32±0.17 |
| SASRec [17] + MIM | 2.67±0.24 | 5.11±0.36 | 3.88±0.24 | 7.03±0.45 | 4.48±0.25 | 8.74±0.30 | 5.66±0.18 | 10.58±0.26 | 7.99±0.17 | 14.77±0.13 | 7.84±0.18 | 14.36±0.33 | 11.31±0.11 | 18.69±0.14 | 12.32±0.14 | 20.67±0.30 |
| BERT4Rec [38] + AMID | 2.99±0.07 | 5.70±0.13 | 3.79±0.15 | 7.09±0.32 | 4.73±0.29 | 9.30±0.41 | 5.70±0.12 | 10.63±0.36 | 7.95±0.23 | 14.49±0.36 | 8.15±0.13 | 14.85±0.28 | 11.26±0.07 | 18.58±0.27 | 12.16±0.09 | 20.08±0.31 |
| GRU4Rec [12] + AMID | 3.10±0.17 | 5.95±0.16 | 3.94±0.15 | 7.30±0.30 | 4.89±0.10 | 9.42±0.28 | 5.90±0.17 | 11.01±0.17 | 8.33±0.09* | 15.49±0.50* | 15.29±0.29* | 15.29±0.29* | 11.74±0.07* | 19.50±0.08* | 20.32±0.17 | 20.85±0.20* |
| SASRec [17] + AMID | 3.20±0.22* | 6.14±0.33* | 4.19±0.10* | 7.62±0.20* | 4.97±0.15* | 9.48±0.22* | 5.96±0.14* | 11.04±0.26* | 8.20±0.10 | 14.99±0.16 | 8.32±0.20 | 14.96±0.15 | 11.71±0.23 | 19.28±0.23 | 12.52±0.13 | 20.79±0.20 |
| Improvement(%) | 26.48 | 24.54 | 10.55 | 7.17 | 17.22 | 11.53 | 5.67 | 4.15 | 9.46 | 10.25 | 4.91 | 7.15 | 8.70 | 8.15 | 2.20 | 2.41 |

"*" denotes statistically significant improvements ($p < 0.05$), as determined by a paired t-test comparison with the second best result in each case.

**Table 2: Experimental results (%) on the bi-directional Loan-Fund and Loan-Account CDSR scenario with different $\mathcal{K}_u$.**

| Methods | Loan-domain recommendation | | | | Fund-domain recommendation | | | | Loan-domain recommendation | | | | Account-domain recommendation | | | |
|---|---|---|---|---|---|---|---|---|---|---|---|---|---|---|---|---|
| | $\mathcal{K}_u$=25% | | $\mathcal{K}_u$=75% | | $\mathcal{K}_u$=25% | | $\mathcal{K}_u$=75% | | $\mathcal{K}_u$=25% | | $\mathcal{K}_u$=75% | | $\mathcal{K}_u$=25% | | $\mathcal{K}_u$=75% | |
| | NDCG@10 | HR@10 | NDCG@10 | HR@10 | NDCG@10 | HR@10 | NDCG@10 | HR@10 | NDCG@10 | HR@10 | NDCG@10 | HR@10 | NDCG@10 | HR@10 | NDCG@10 | HR@10 |
| BERT4Rec [38] | 33.12±0.26 | 41.19±0.29 | 35.76±0.38 | 45.55±0.36 | 21.73±0.58 | 32.45±0.86 | 21.56±0.79 | 34.13±0.96 | 27.49±0.21 | 37.15±0.25 | 27.89±0.70 | 40.03±0.22 | 32.84±0.50 | 43.16±0.33 | 34.05±0.49 | 45.49±0.15 |
| GRU4Rec [12] | 35.01±0.19 | 44.15±0.27 | 36.04±0.19 | 46.25±0.13 | 25.94±0.51 | 38.26±0.24 | 26.55±0.32 | 38.57±0.27 | 29.01±0.68 | 40.48±0.22 | 29.01±0.51 | 41.43±0.16 | 34.17±0.28 | 45.32±0.18 | 34.28±0.15 | 46.07±0.22 |
| SASRec [17] | 35.02±0.36 | 44.03±0.44 | 36.06±0.18 | 46.24±0.15 | 26.07±0.34 | 38.47±0.58 | 27.29±0.55 | 39.68±0.38 | 29.47±0.41 | 40.42±0.25 | 29.81±0.85 | 41.15±0.45 | 34.16±0.59 | 45.33±0.33 | 34.22±0.32 | 46.09±0.14 |
| STAR [35] | 34.35±0.17 | 43.07±0.28 | 35.64±0.19 | 45.75±0.33 | 26.17±0.52 | 38.15±0.37 | 26.71±0.30 | 38.70±0.36 | 29.06±0.49 | 39.58±0.24 | 29.06±0.36 | 40.97±0.20 | 34.34±0.45 | 44.77±0.13 | 34.35±0.21 | 45.89±0.11 |
| MAMDR [25] | 34.38±0.14 | 43.15±0.26 | 35.72±0.17 | 45.88±0.31 | 26.20±0.49 | 38.28±0.35 | 26.80±0.28 | 38.78±0.32 | 29.08±0.46 | 39.66±0.22 | 29.12±0.39 | 41.06±0.23 | 34.39±0.30 | 44.80±0.21 | 34.40±0.14 | 46.03±0.14 |
| SSCDR [16] | 34.41±0.11 | 43.23±0.23 | 35.62±0.10 | 45.80±0.32 | 26.27±0.40 | 38.19±0.54 | 26.82±0.41 | 38.91±0.42 | 29.12±0.43 | 39.70±0.33 | 29.11±0.40 | 41.00±0.51 | 34.21±0.25 | 44.72±0.60 | 34.37±0.55 | 46.16±0.34 |
| Pi-Net [27] | 34.76±0.33 | 43.63±0.36 | 36.01±0.22 | 45.91±0.18 | 24.66±1.48 | 37.23±0.55 | 24.37±0.72 | 37.58±0.66 | 28.98±0.34 | 39.86±0.47 | 29.09±0.51 | 41.24±0.30 | 34.36±0.24 | 45.26±0.12 | 34.33±0.40 | 46.15±0.20 |
| DASL [19] | 35.16±0.35 | 44.34±0.42 | 35.96±0.17 | 46.22±0.08 | 26.44±0.57 | 38.35±0.52 | 25.55±0.51 | 38.59±0.39 | 29.09±0.44 | 40.31±0.51 | 29.16±0.36 | 41.4s3±0.25 | 34.69±0.31 | 45.36±0.31 | 34.55±0.13 | 46.07±0.22 |
| C$^2$DSR [2] | 34.78±0.44 | 43.52±0.46 | 35.99±0.27 | 46.15±0.21 | 23.54±0.37 | 36.34±0.73 | 26.57±0.39 | 38.42±0.41 | 29.28±0.64 | 40.18±0.39 | 29.08±0.31 | 41.45±0.29 | 33.54±0.27 | 45.37±0.17 | 34.02±0.39 | 46.42±0.17 |
| DCRec [47] | 33.43±0.21 | 41.97±0.57 | 35.44±0.35 | 45.58±0.41 | 21.21±0.79 | 34.73±0.39 | 24.41±0.40 | 37.22±0.51 | 27.51±0.39 | 38.41±0.43 | 28.60±0.48 | 40.65±0.58 | 32.99±0.39 | 43.89±0.19 | 33.81±0.21 | 45.69±0.31 |
| BERT4Rec [38] + CaseQ [46] | 35.29±0.16 | 44.25±0.20 | 35.81±0.09 | 45.96±0.14 | 26.26±0.43 | 38.54±0.26 | 27.38±0.19 | 39.20±0.22 | 29.49±0.18 | 40.46±0.09 | 30.37±0.11 | 41.32±0.14 | 34.77±0.13 | 45.25±0.26 | 34.62±0.11 | 46.43±0.08 |
| GRU4Rec [12] + CaseQ [46] | 35.35±0.11 | 44.11±0.20 | 35.82±0.06 | 45.94±0.09 | 26.47±0.51 | 38.71±0.36 | 27.35±0.16 | 39.26±0.14 | 29.42±0.11 | 40.47±0.15 | 30.39±0.19 | 41.44±0.25 | 34.77±0.14 | 45.18±0.28 | 34.44±0.11 | 46.39±0.18 |
| SASRec [17] + CaseQ [46] | 35.37±0.14 | 44.21±0.20 | 35.83±0.10 | 45.89±0.18 | 26.44±0.30 | 38.53±0.25 | 27.48±0.12 | 39.22±0.13 | 29.41±0.12 | 40.59±0.19 | 30.35±0.18 | 41.39±0.16 | 34.84±0.12 | 45.47±0.15 | 34.56±0.18 | 46.34±0.20 |
| BERT4Rec [38] + IPSCDR [21] | 33.56±0.34 | 42.05±0.47 | 35.56±0.29 | 45.70±0.25 | 21.39±0.86 | 34.82±0.49 | 24.57±0.33 | 37.34±0.34 | 27.68±0.31 | 38.35±0.41 | 28.58±0.44 | 40.63±0.41 | 32.92±0.46 | 43.96±0.25 | 33.76±0.25 | 45.67±0.20 |
| GRU4Rec [12] + IPSCDR [21] | 34.37±0.16 | 42.99±0.28 | 35.79±0.15 | 46.00±0.15 | 26.45±0.57 | 38.43±0.39 | 26.95±0.62 | 39.14±0.36 | 28.76±0.48 | 40.04±0.48 | 28.64±0.29 | 41.07±0.17 | 33.92±0.50 | 45.27±0.16 | 34.21±0.20 | 46.15±0.13 |
| SASRec [17] + IPSCDR [21] | 34.42±0.35 | 43.08±0.37 | 36.03±0.25 | 46.43±0.07 | 26.24±0.31 | 38.65±0.29 | 28.05±0.28 | 39.89±0.17 | 29.12±0.47 | 40.07±0.20 | 30.08±0.24 | 41.40±0.07 | 33.83±0.26 | 45.10±0.26 | 34.27±0.12 | 46.41±0.12 |
| BERT4Rec [38] + MIM | 33.47±0.45 | 41.45±0.49 | 36.10±0.24 | 46.02±0.24 | 21.81±0.95 | 33.42±0.90 | 22.71±1.12 | 35.23±0.58 | 27.76±0.50 | 37.92±0.25 | 28.97±0.68 | 40.83±0.11 | 33.30±0.48 | 43.81±0.43 | 34.25±0.48 | 45.73±0.15 |
| GRU4Rec [12] + MIM | 35.50±0.12 | 44.59±0.18 | 36.42±0.21 | 46.56±0.10 | 26.48±0.91 | 38.54±0.51 | 26.97±0.72 | 39.13±0.48 | 29.28±0.64 | 40.74±0.22 | 29.28±0.48 | 41.82±0.17 | 34.96±0.40 | 46.01±0.19 | 34.86±0.44 | 46.51±0.19 |
| SASRec [17] + MIM | 35.40±0.31 | 44.31±0.47 | 36.48±0.16 | 46.54±0.08 | 26.73±0.36 | 39.00±0.12 | 28.15±0.36 | 40.30±0.38 | 29.98±0.46 | 40.66±0.24 | 30.54±0.49 | 41.48±0.34 | 34.61±0.32 | 45.75±0.08 | 34.64±0.55 | 46.47±0.17 |
| BERT4Rec [38] + AMID | 36.55±0.26 | 46.57±0.15 | 36.79±0.29* | 46.88±0.08* | 27.43±0.35 | 38.79±0.94 | 27.10±1.28 | 38.82±1.03 | 30.69±0.98 | 41.42±0.40 | 30.21±1.37 | 41.42±0.22 | 35.57±0.95 | 46.60±0.20 | 35.17±0.62 | 46.46±0.08 |
| GRU4Rec [12] + AMID | 36.53±0.17 | 46.54±0.10 | 36.56±0.11 | 46.85±0.13 | 27.52±0.41 | 39.38±0.26 | 27.05±0.47 | 39.36±0.38 | 30.78±0.51 | 41.76±0.31 | 30.62±0.52 | 42.08±0.16 | 35.66±0.22 | 46.71±0.08 | 35.38±0.47 | 46.64±0.04 |
| SASRec [17] + AMID | 36.76±0.09* | 46.73±0.10* | 36.58±0.12 | 46.76±0.11 | 27.79±0.49* | 39.92±0.47* | 28.48±0.51* | 40.54±0.20* | 31.64±0.46* | 42.01±0.17* | 32.24±0.39* | 42.45±0.24* | 36.30±0.21* | 46.86±0.18* | 36.66±0.11* | 47.13±0.17* |
| Improvement(%) | 3.93 | 5.60 | 2.11 | 0.97 | 4.99 | 3.13 | 1.53 | 1.63 | 7.29 | 3.50 | 6.09 | 2.44 | 4.19 | 3.06 | 6.08 | 1.51 |

users across domains, with a higher $\mathcal{K}_u$ indicating a less biased environment. Following previous CDSR literature [2, 19, 27], we employed the leave-one-out technique to evaluate the performance of our developed model. In order to ensure an unbiased evaluation, we adopt the methodology employed in previous works [18, 51], wherein we randomly select 999 negative items (i.e., items that the user has not interacted with) and combine them with 1 positive item (i.e., a ground-truth interaction) to form our recommendation candidates for the ranking test. We evaluated our model using the normalized discounted cumulative gain (NDCG@10) and hit rate (HR@10) metrics, which are common in CDSR literature. For all comparative models, we ran each experiment five times and reported the results by mean and variance. Further details about our experimental setup are in Appendix C.1.

**Table 3: Statistics on the Amazon datasets.**

| Dataset | | Users | Items | Ratings | #Overlap | Avg.length | Density |
|---|---|---|---|---|---|---|---|
| Amazon | Cloth | 27,519 | 9,481 | 161,010 | 16,337 | 4.39 | 0.06% |
| | Sport | 107,984 | 40,460 | 851,553 | | 7.58 | 0.02% |
| Amazon | Phone | 41,829 | 17,943 | 194,121 | 7,857 | 4.53 | 0.03% |
| | Elec | 27,328 | 12,655 | 170,426 | | 6.19 | 0.05% |
| MYbank | Loan | 39,557,003 | 61,934 | 227,079,281 | 29,476,198 | 1.82 | 0.01% |
| | Fund | 48,439,382 | 13,927 | 133,836,385 | | 2.57 | 0.02% |
| MYbank | Loan | 39,557,003 | 61,934 | 227,079,281 | 37,821,145 | 1.82 | 0.01% |
| | Account | 92,692,975 | 21,599 | 278,948,331 | | 2.31 | 0.01% |

#Overlap: the number of overlapping users across domains.

## 5.2 Performance Comparisons

**Compared Methods.** We compare our method with three classes of baselines: (1) Single-domain sequential recommendation methods, i.e., BERT4Rec [38], GRU4Rec [12] and SASRec [17]. (2) Conventional Cross-domain recommendation methods, i.e., STAR [35], MAMDR [25], SSCDR [16]. (3) Cross-domain sequential recommendation methods, i.e., Pi-Net [27], DASL [19] and C$^2$DSR [2]. (4) Debiased recommendation methods, i.e., DCRec [47], CaseQ [46] and IPSCDR [21]. A detailed introduction to these baselines can be found in Appendix C.2. As shown in Table 4, our AMID is the most versatile and universal approach which considers propagating cross-domain knowledge for both overlapping users and non-overlapping users and can combine with most off-the-shelf SDSR backbone models. Regarding the debiasing frameworks [46, 47], our AMID approach can simultaneously alleviate multiple types of bias, especially selection bias, from multiple domains. Additionally, our proposed doubly robust estimator for CDSR has a low variance, which is different from the IPS estimator from IPSCDR [21] with high variance [8].

Table 4: Comparisons with existing baselines along different dimensions.

| Method | Cross-domain Scenario | | | | Debiasing Framework | | |
|---|---|---|---|---|---|---|---|
| | Fully | Partially | Non-overlap | Universal | Domain | Variance | Universal |
| Pi-Net [27] | ✓ | - | - | - | - | - | - |
| DASL [19] | ✓ | ✓ | - | - | - | - | - |
| C$^2$DSR [2] | ✓ | ✓ | - | - | - | - | - |
| STAR [35] | ✓ | ✓ | - | - | - | - | - |
| MAMDR [25] | ✓ | ✓ | - | ✓ | - | - | - |
| SSCDR [16] | ✓ | ✓ | - | - | - | - | - |
| DCRec [47] | - | - | - | - | Single | - | - |
| CaseQ [46] | - | - | - | - | Single | - | ✓ |
| IPSCDR [21] | ✓ | ✓ | - | ✓ | Multiple | High | ✓ |
| **AMID(Ours)** | ✓ | ✓ | ✓ | ✓ | Multiple | **Low** | ✓ |

**Quantitative Results.** Tables 1–2 present the quantitative comparison results on two CDSR tasks with different selection bias magnitudes. A larger $\mathcal{K}_u$ indicates a less biased scenario. The best results of each column are highlighted in boldface, while the second-best results are underlined. As expected, the performance of all models increases with increasing $\mathcal{K}_u$, since more biased scenarios may make it harder for models to converge. Our analysis yields the following insightful findings: (1) In most cases, the debiasing baselines, which eliminate the biases produced in domains, perform better than the CDSR baselines in more biased scenarios (i.e. $\mathcal{K}_u = 25\%$). (2) In a more biased scenario with smaller $\mathcal{K}_u$, our framework achieves more significant performance compared to the second-best models, indicating that our AMID effectively alleviates the bias. (3) Benefitting from the low variance in our estimator, our AMID is more unbiased and performs better than IPSCDR.

**Model Efficiency.** All comparative models are trained and tested on the same machine, which has a single NVIDIA GeForce A100 with 80GB memory and an Intel Core i7-8700K CPU with 64G RAM. Notably, the number of parameters for typical C$^2$DSR, SASRec + CaseQ, SASRec + IPSCDR, and SASRec + AMID were of the same order of magnitude, denoted as 0.276M, 0.210M, 0.192M, and 0.193M. The training/testing efficiencies of C$^2$DSR, SASRec + CaseQ, SASRec + IPSCDR, and SASRec + AMID in processing one batch of samples are 0.130s/0.049s, 0.083s/0.027s, 0.143s/0.045s, and 0.111s/0.032s, respectively. Therefore, our AMID achieves superior performance

enhancements in open-world CDSR scenarios while maintaining promising time efficiency.

**Ablation Study.** To better evaluate the effectiveness of each key component in our approach, we conducted an ablation study by comparing it with a variant that only utilized MIM. Notably, we did not include a variant that only employed DRE, as our approach would degrade into an SDSR method in the absence of MIM. However, our variant equipped with only the multi-interest information module (MIM) still achieves state-of-the-art results in most cases. This is because our proposed module can effectively propagate potential interest information among both overlapping and non-overlapping users.

## 5.3 Online A/B Test

We conduct large-scale online A/B tests on open-world financial CDSR scenarios with partially overlapping users. In the online serving platform, a large number of users participate in one or multiple financial domains, such as purchasing funds, mortgage loans, or discounting bills. Specifically, we selected three popular domains - "Loan," "Fund," and "Account" - from the serving platform, with partially overlapping users, as the targets of our online testing. We calculate the average statistics of online traffic logs for one day and present them in Table 5. For the control group, we adopt the current online solution for recommending themes to users, which is a cross-domain sequential recommendation method that utilizes noisy auxiliary behaviors directly. For the experiment group, we equip our method with a mature SDSR approach that has achieved remarkable success in the business. We evaluate the results based on three metrics: the number of users who have been exposed to the service, the number of users who have clicked inside the service, and the conversion rate of the service (denoted by # exposure, # clk, and CVR, respectively). All of the results are reported as the lift compared to the control group and presented in Table 6. In a fourteen-day online A/B test, our method improved the average exposure by 9.65%, the click rate by 5.69%, and the CVR by 1.32% in the three domains.

Table 5: Average statistics of online traffic logs for 1 day.

| Dataset | Users | Items | Ratings | #Overlap | Density |
|---|---|---|---|---|---|
| Loan | 14,345,278 | 41,156 | 683,125,459 | | 0.12% |
| Fund | 1,219,254 | 3,104 | 1,194,405 | 1,109,493 | 0.03% |
| Account | 3,461,290 | 8,925 | 9,147,610 | | 0.03% |

Table 6: Online A/B testing results from 9.15 to 9.28, 2023

| | # exposure | # clk | CVR |
|---|---|---|---|
| Loan Domain | +9.89% | +5.27% | +1.42% |
| Fund Domain | +7.32% | +4.94% | +0.98% |
| Account Domain | +11.73% | +6.85% | +1.57% |

## 5.4 Hyperparameter Analysis

**The threshold $k$ for the group.** We conduct ablation experiments by varying the threshold $k \in \{0.5, 0.6, 0.7, 0.8, 0.9\}$ to investigate the impact of the threshold $k$ on constructing interest groups in the multi-interest information module. Our results show that a

larger threshold $k$ ($0.5 \rightarrow 0.7$) leads to better performance, as more related interest information can be transferred. However, when the threshold $k$ is increased beyond 0.7, the model's performance drops due to noise interference and redundant information. Therefore, to achieve superior performance, we set the threshold $k$ to 0.7.

**The number of the sampled users.** To explore the impact of the number of sampled users on the multi-interest information module, we conduct ablation experiments varying the number of sampled users from 128 to 1024. Our findings suggest that an increase in the number of sampled users initially improves the recommendation performance, but it eventually declines when the matching neighbors reach 1024. This observation indicates that having too few sampled users would provide insufficient transferred information, while too many sampled users could introduce interference noise and compromise the model's performance. In practice, we select the number of sampled users to be 512 as it results in the best performance for our model. More analysis and results can be found in Appendix C.3.

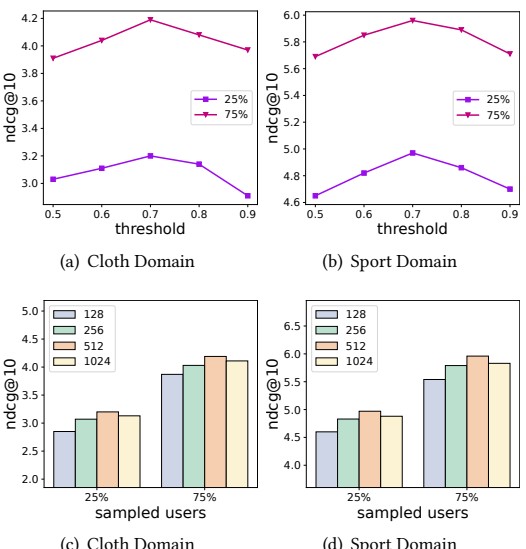

(a) Cloth Domain     (b) Sport Domain

(c) Cloth Domain     (d) Sport Domain

**Figure 6: Impact of the threshold $k$ and the number of the sampled users.**

## 6 RELATED WORK

**Conventional cross-domain recommendation** has emerged as a promising solution for mitigating data sparsity and cold-start issues encountered in single-domain recommendation systems. Early CDR studies [14, 23] have primarily focused on developing approaches that transfer cross-domain knowledge by relying on overlapping users. However, real-world CDR scenarios often do not satisfy strict overlapping requirements and exhibit only a small fraction of common users across domains. To tackle this challenge, recent methods [25, 35]) have proposed a network structure comprising shared and domain-specific networks to effectively capture the unique characteristics and commonalities across all domains simultaneously. While these CDR approaches incorporate valuable information from relevant domains to enhance performance in the target domain, they still encounter difficulties in addressing the contextual

sequential dependencies within users' interaction history, which are essential for comprehensive modeling in CDSR tasks.

**Cross-domain sequential recommendation** is designed to improve recommendations for SR tasks that involve items from multiple domains. Pi-Net [27] and PSJNet [39] devise the gating mechanisms to transfer the information among the overlapping users. Similarly, DASL [19] designs a dual-attention mechanism to bidirectionally transfer user preferences within the overlapping users. The interaction bipartite graph [9, 26] is constructed to propagate the information among users. $C^2DSR$ [2] introduces a contrastive objective combined with GNNs to enhance the representation of user preferences. However, these works construct their cross-domain unit relying on the overlapping users under closed-world assumptions, leading to worse performance in the open-world case.

**Debias for recommender systems** are proposed to alleviate the widespread bias in the user behavior observed data, including the selection bias [5, 10], position bias [13, 15], exposure bias [22, 33] and popularity bias [1, 46]. To address selection bias in RS, two standard approaches have been proposed: the error-imputation-based (EIB) approach and the inverse-propensity-scoring (IPS) approach. The EIB approach [37] estimates the prediction error for missing ratings, while the IPS approach [34, 43] reduces selection bias by optimizing the risk function with the inverse propensity score. Recently, [21] proposes an IPS estimator for cross-domain scenarios with multiple restrictions. However, EIB methods may have a large bias due to imputation inaccuracy [7], and propensity-based methods may suffer from high variance [32], leading to non-optimal results. [40] propose a self-normalized inverse propensity scoring estimator to reduce the variance of the IPS estimator. To design a less biased estimator, the doubly robust model [42] integrates the imputation model with the propensity score for the single-domain recommendation.

## 7 CONCLUSIONS AND DISCUSSIONS

In this paper, we conduct a thorough study of existing CDSR methods under open-world assumptions. To overcome the challenges, we devise an adaptive multi-interest debiasing framework that includes a multi-interest information module (MIM) and a doubly robust estimator (DRE) for CDSR. MIM utilizes the user behaviors to build the interest groups and propagate the information among both overlapping and non-overlapping users, while DRE introduces a cross-domain debiasing estimator to reduce the estimation bias in an open-world environment. Besides, we collect a financial industry dataset from Alipay, which includes over one billion users. Extensive offline and online experiments show the remarkable efficacy of our approach, as it outperforms existing methods including CDSR methods and debiasing methods in various evaluation metrics.

**Limitations.** Our MIM constructs interest groups between pairs of domains, which share cross-domain knowledge as widely as possible. In the real-world platform, commercial activities are usually composed of multiple domains. However, constructing all the groups for $|\mathcal{Z}|$ domains has a time complexity of $O(|\mathcal{Z}|^2)$, which can become quite time-consuming as the number of domains increases. Therefore, it is important to develop more efficient methods for constructing groups among multiple domains in future work.

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

# A APPENDIX A: PROOFS OF LEMMAS AND THEOREMS

**Lemma 4.1 (Bias of DR Estimator)**. Given imputation errors $\hat{\mathbf{E}}^Z$ and learned propensities $\hat{\mathbf{P}}^Z$ for all user-item pairs, the bias of the DR estimator in the CDSR task is

$$\text{Bias}(\mathcal{E}_{DR}^*) = \frac{1}{|\mathcal{Z}|} \sum_{Z \in \mathcal{Z}} \left[ \frac{1}{|\mathcal{D}^Z|} \left| \sum_{u,v \in \mathcal{D}^Z} \Delta_{u,v}^Z \delta_{u,v}^Z \right| \right] \quad (17)$$

PROOF. According to the definition of the bias, we can derive the bias of the DR estimator for CDSR as follows.

$$\text{Bias}(\mathcal{E}_{DR}^*) \quad (18)$$
$$= |\mathcal{P} - \mathbb{E}_{\mathbf{O}}[\mathcal{E}_{DR}^*]|, \quad (19)$$
$$= \left| \frac{1}{|\mathcal{Z}|} \sum_{Z \in \mathcal{Z}} \left[ \frac{1}{|\mathcal{D}^Z|} \sum_{u,v \in \mathcal{D}^Z} \left( e_{u,v}^Z - \hat{e}_{u,v}^Z - \frac{p_{u,v}^Z \delta_{u,v}^Z}{\hat{p}_{u,v}^Z} \right) \right] \right|, \quad (20)$$
$$= \left| \frac{1}{|\mathcal{Z}|} \sum_{Z \in \mathcal{Z}} \left[ \frac{1}{|\mathcal{D}^Z|} \sum_{u,v \in \mathcal{D}^Z} \left( \delta_{u,v}^Z - \frac{p_{u,v}^Z \delta_{u,v}^Z}{\hat{p}_{u,v}^Z} \right) \right] \right|, \quad (21)$$
$$= \left| \frac{1}{|\mathcal{Z}|} \sum_{Z \in \mathcal{Z}} \left[ \frac{1}{|\mathcal{D}^Z|} \sum_{u,v \in \mathcal{D}^Z} \left( \frac{(\hat{p}_{u,v}^Z - p_{u,v}^Z) \delta_{u,v}^Z}{\hat{p}_{u,v}^Z} \right) \right] \right|, \quad (22)$$
$$= \frac{1}{|\mathcal{Z}|} \sum_{Z \in \mathcal{Z}} \left[ \frac{1}{|\mathcal{D}^Z|} \left| \sum_{u,v \in \mathcal{D}^Z} \Delta_{u,v}^Z \delta_{u,v}^Z \right| \right], \quad (23)$$
$$\quad (24)$$

which completes the proof. Following the previous works [36, 42], the imputation error $\delta_{u,v}^Z$ and the learned propensities $\Delta_{u,v}^Z$ is defined as:

$$\delta_{u,v}^Z = e_{u,v}^Z - \hat{e}_{u,v}^Z, \quad \Delta_{u,v}^Z = \frac{\hat{p}_{u,v}^Z - p_{u,v}^Z}{\hat{p}_{u,v}^Z} \quad (25)$$

□

**Corollary 4.1 (Double Robustness)**. The DR estimator for CDSR is unbiased when either imputed errors $\hat{\mathbf{E}}^Z$ or learned propensities $\hat{\mathbf{P}}^Z$ are accurate for all user-item pairs.

PROOF. In one respect, when the imputation error is accurate, we have $\delta_{u,v}^Z = 0$ for $u, v \in \mathcal{D}^Z$. In this case, the bias of the DR estimator for CDSR is computed by

$$\text{Bias}(\mathcal{E}_{DR}^*) \quad (26)$$
$$= \frac{1}{|\mathcal{Z}|} \sum_{Z \in \mathcal{Z}} \left[ \frac{1}{|\mathcal{D}^Z|} \left| \sum_{u,v \in \mathcal{D}^Z} \Delta_{u,v}^Z \delta_{u,v}^Z \right| \right], \quad (27)$$
$$= \frac{1}{|\mathcal{Z}|} \sum_{Z \in \mathcal{Z}} \left[ \frac{1}{|\mathcal{D}^Z|} \left| \sum_{u,v \in \mathcal{D}^Z} \Delta_{u,v}^Z \cdot 0 \right| \right], \quad (28)$$
$$= 0. \quad (29)$$

In the other respect, when the learned propensities are accurate, we have $\Delta_{u,v}^Z = 0$ for $u, v \in \mathcal{D}^Z$. In such case, we can compute the bias of the DR estimator for CDSR by

$$\text{Bias}(\mathcal{E}_{DR}^*) \quad (30)$$
$$= \frac{1}{|\mathcal{Z}|} \sum_{Z \in \mathcal{Z}} \left[ \frac{1}{|\mathcal{D}^Z|} \left| \sum_{u,v \in \mathcal{D}^Z} \Delta_{u,v}^Z \delta_{u,v}^Z \right| \right], \quad (31)$$
$$= \frac{1}{|\mathcal{Z}|} \sum_{Z \in \mathcal{Z}} \left[ \frac{1}{|\mathcal{D}^Z|} \left| \sum_{u,v \in \mathcal{D}^Z} 0 \cdot \delta_{u,v}^Z \right| \right], \quad (32)$$
$$= 0. \quad (33)$$

When either imputed errors or learned propensities are accurate, the bias of the proposed estimator is accurate. The proof is completed.

□

**Lemma 4.2 (Tail Bound of DR Estimator)**. Given imputation errors $\hat{\mathbf{E}}^Z$ and learned propensities $\hat{\mathbf{P}}^Z$, for any prediction matrix $\hat{\mathbf{R}}^Z$, with probability 1-$\eta$, the deviation of the DR estimator from its expectation has the following tail bound in CDSR task.

$$\left| \mathcal{E}_{DR}^* - \mathbb{E}_{\mathbf{O}}[\mathcal{E}_{DR}^*] \right| \leq \quad (34)$$
$$\sqrt{\frac{\log(\frac{2}{\eta})}{2|\mathcal{Z}|(\sum_{Z \in \mathcal{Z}} |\mathcal{D}^Z|)^2} \sum_{Z \in \mathcal{Z}} \left[ \frac{1}{|\mathcal{D}^Z|} \sum_{u,v \in \mathcal{D}^Z} \left( \frac{\delta_{u,v}^Z}{\hat{p}_{u,v}^Z} \right)^2 \right]} \quad (35)$$

PROOF. To avoid cluttering the notation, we introduce a random variable $\mathsf{X}_{u,v}^Z$ denoted by

$$\mathsf{X}_{u,v}^Z = \hat{e}_{u,v}^Z + \frac{o_{u,v}^Z \delta_{u,v}^Z}{\hat{p}_{u,v}^Z} \quad (36)$$

Considering the observation indicator $o_{u,v}^Z$ follows a Bernoulli distribution with probability $p_{u,v}^Z$, we can obtain the distribution pattern of the random variable $\mathsf{X}_{u,v}^Z$ as follows.

$$P(\mathsf{X}_{u,v}^Z) = \begin{cases} p_{u,v}^Z, & \mathsf{X}_{u,v}^Z = \hat{e}_{u,v}^Z + \kappa_{u,v}^Z \\ 1 - p_{u,v}^Z, & \mathsf{X}_{u,v}^Z = \hat{e}_{u,v}^Z \end{cases} \quad (37)$$

where $\kappa_{u,v}^Z$ is given by

$$\kappa_{u,v}^Z = \frac{e_{u,v}^Z - \hat{e}_{u,v}^Z}{\hat{p}_{u,v}^Z} = \frac{\delta_{u,v}^Z}{\hat{p}_{u,v}^Z} \quad (38)$$

Determining the interval $[\hat{e}^Z_{u,v}, \hat{e}^Z_{u,v} + \kappa^Z_{u,v}]$ for the random variable $\kappa^Z_{u,v}$ of size $\kappa^Z_{u,v}$ with probability 1 is a simple process. This is facilitated by the assumption that the observation indicators $o^Z_{u,v}$ are independent random variables, which ensures that the random variables $\kappa^Z_{u,v}$ are also independent. The general form of Hoeffding's inequality for bounded random variables [41] can be expressed as 39. Let $X_1, ..., X_N$ be independent random variables. For each $i$, we assume that $X_i \in [a_i, b_i]$. For any $\epsilon > 0$, we have the inequality

$$P\left(\left|\sum_{i=1}^{N} X_i - \sum_{i=1}^{N} \mathbb{E}(X_i)\right| \geq \epsilon\right) \leq 2\exp\left(\frac{-2\epsilon^2 N^2}{\sum_{i=1}^{N}(b_i - a_i)^2}\right) \quad (39)$$

According to Hoeffding's inequality, for any $\epsilon > 0$, we have the following inequality

$$P\left(\left|\frac{1}{|\mathcal{Z}|}\sum_{Z\in\mathcal{Z}}\frac{1}{|\mathcal{D}^Z|}\left[\sum_{u,v\in\mathcal{D}^Z}\chi^Z_{u,v}\right] - \frac{1}{|\mathcal{Z}|}\sum_{Z\in\mathcal{Z}}\frac{1}{|\mathcal{D}^Z|}\left[\sum_{u,v\in\mathcal{D}^Z}\mathbb{E}_{\mathbf{O}}(\chi^Z_{u,v})\right]\right| \geq \epsilon\right)$$
$$(40)$$

$$\leq 2\exp\left(\frac{-2\epsilon^2\left(\sum_{Z\in\mathcal{Z}}|D^Z|\right)^2}{\frac{1}{|\mathcal{Z}|}\sum_{Z\in\mathcal{Z}}\frac{1}{|\mathcal{D}^Z|}\sum_{u,v\in\mathcal{D}^Z}(\kappa^Z_{u,v})^2}\right) \quad (41)$$

To solve for $\epsilon$, one can set the right side of the inequality to be $\eta$ and proceed with the following steps.

$$\eta = 2\exp\left(\frac{-2\epsilon^2\left(\sum_{Z\in\mathcal{Z}}|D^Z|\right)^2}{\frac{1}{|\mathcal{Z}|}\sum_{Z\in\mathcal{Z}}\frac{1}{|\mathcal{D}^Z|}\sum_{u,v\in\mathcal{D}^Z}(\kappa^Z_{u,v})^2}\right) \quad (42)$$

$$\iff \log(\frac{\eta}{2}) = \frac{-2\epsilon^2\left(\sum_{Z\in\mathcal{Z}}|D^Z|\right)^2}{\frac{1}{|\mathcal{Z}|}\sum_{Z\in\mathcal{Z}}\frac{1}{|\mathcal{D}^Z|}\sum_{u,v\in\mathcal{D}^Z}(\kappa^Z_{u,v})^2} \quad (43)$$

$$\iff \epsilon = \sqrt{\frac{\log(\frac{2}{\eta})}{2|\mathcal{Z}|(\sum_{Z\in\mathcal{Z}}|D^Z|)^2}\sum_{Z\in\mathcal{Z}}\left[\frac{1}{|\mathcal{D}^Z|}\sum_{u,v\in\mathcal{D}^Z}(\kappa^Z_{u,v})^2\right]} \quad (44)$$

$$\iff \epsilon = \sqrt{\frac{\log(\frac{2}{\eta})}{2|\mathcal{Z}|(\sum_{Z\in\mathcal{Z}}|D^Z|)^2}\sum_{Z\in\mathcal{Z}}\left[\frac{1}{|\mathcal{D}^Z|}\sum_{u,v\in\mathcal{D}^Z}(\frac{\delta^Z_{u,v}}{\hat{p}^Z_{u,v}})^2\right]} \quad (45)$$

The proof is completed.                                    □

**Corollary 4.2 (Tail Bound Comparison).** Suppose imputed errors $\hat{\mathbf{E}}^Z$ are such that $0 \leq \hat{e}^Z_{u,v} \leq 2e^Z_{u,v}$ for each $u, v \in \mathcal{D}^Z$, then for any learned propensities $\hat{\mathbf{P}}$, the tail bound of the proposed estimator will be lower than that of the IPS estimator which is utilized in IPSCDR [21].

PROOF. We can derive the following inequalities

$$0 \leq \hat{e}^Z_{u,v} \leq 2e^Z_{u,v} \text{ for } Z \in \mathcal{Z} \text{ for } u, v \in \mathcal{D}^Z \quad (46)$$

$$\implies \hat{e}^Z_{u,v} - e^Z_{u,v} \leq e^Z_{u,v} \quad (47)$$

$$\implies (\delta^Z_{u,v})^2 \leq (e^Z_{u,v})^2 \quad (48)$$

$$\implies \sqrt{\left(\frac{\delta^Z_{u,v}}{\hat{p}^Z_{u,v}}\right)^2} \leq \sqrt{\left(\frac{e^Z_{u,v}}{\hat{p}^Z_{u,v}}\right)^2} \quad (49)$$

$$\implies \sqrt{\frac{\log(\frac{2}{\eta})}{2|\mathcal{Z}|(\sum_{Z\in\mathcal{Z}}|D^Z|)^2}\sum_{Z\in\mathcal{Z}}\left[\frac{1}{|\mathcal{D}^Z|}\sum_{u,v\in\mathcal{D}^Z}(\frac{\delta^Z_{u,v}}{\hat{p}^Z_{u,v}})^2\right]} \quad (50)$$

$$\leq \sqrt{\frac{\log(\frac{2}{\eta})}{2|\mathcal{Z}|(\sum_{Z\in\mathcal{Z}}|D^Z|)^2}\sum_{Z\in\mathcal{Z}}\left[\frac{1}{|\mathcal{D}^Z|}\sum_{u,v\in\mathcal{D}^Z}(\frac{e^Z_{u,v}}{\hat{p}^Z_{u,v}})^2\right]} \quad (51)$$

□

In the last inequality, the first row denotes the tail bound of the DR estimator for CDSR and the second row denotes the tail bound of the IPS estimator for CDSR [21, 34]. This completes the proof.

# B  APPENDIX B: METHODOLOGY

Following the previous work [42], we also adopt a joint learning mechanism for training our model. The optimization procedures are shown in Alg. 1.

---

**Algorithm 1** The optimization scheme of AMID.

---

**Input:** The true ratings $\mathbf{R}$, the learned propensities $\hat{\mathbf{P}}$ from observed data $O$.
1: **for** $step$ $q \in \{1, ..., Q\}$ **do**
2:     Sample a batch of user-item pairs in multiple domains $\mathcal{Z}$ from $O$.
3:     Compute the loss function $\mathcal{L}_e$.
4:     Update the model parameter $\theta^{q+1} \leftarrow \theta^q - \eta\nabla_\theta\mathcal{L}_e(\theta, \phi, \psi)$
5:     Update the propensities model parameter $\phi^{q+1} \leftarrow \phi^q - \eta\nabla_\phi\mathcal{L}_e(\theta, \phi, \psi)$
6:     Update the imputation model parameter $\psi^{q+1} \leftarrow \psi^q - \eta\nabla_\psi\mathcal{L}_e(\theta, \phi, \psi)$
7: **end for**
8: **for** $step$ $q \in \{1, ..., Q'\}$ **do**
9:     Sample a batch of user-item pairs in multiple domains $\mathcal{Z}$ from $\mathcal{D}$.
10:     Compute the loss function $\mathcal{L}_r$.
11:     Update the model parameter $\theta^{q+1} \leftarrow \theta^q - \eta'\nabla_\theta\mathcal{L}_r(\theta, \phi, \psi)$
12: **end for**

---

# C  APPENDIX C: EXPERIMENTS

## C.1  Experiment Setup

**Evaluation Metrics** In each domain, we divided the users into three sets: 80% for training, 10% for validation, and 10% for testing. To preprocess the data, items with fewer than 10 interactions and users with fewer than 5 interactions in their respective domains were filtered out, following the approach of previous studies [3, 45].

This ensured that the embeddings learned by the users/items were representative of their source domain. A non-overlapping ratio $\mathcal{K}_u$ was introduced to control the number of non-overlapping users and simulate different debiased scenarios. For example, in the Amazon "Cloth-Sport" dataset with $\mathcal{K}_u$ = 25%, the number of overlapped users in the training set was calculated as (27,519 + 107,984 - 16,377 * 2) * 0.25 * 0.8 = 20,549. The same sampling strategy was applied to the validation set, while the test set was not downsampled. This sampling strategy can simulate the occurrence of selection bias in the open-world environment, where the training set mainly consists of users who are more likely to be selected or exposed, while the test set covers a wider range of users without careful selection. All evaluation metrics used in this study indicate better performance with higher values. Regarding the unseen users in $\mathcal{D}$, we use the non-overlapping users who were not selected for the observed data as a substitute for unseen users. We remove the actual ratings for the unseen users while retaining their sequences.

**Parameter Settings** To ensure a fair comparison between different approaches, we set the same hyper-parameters for all of them. Specifically, we fixed the embedding dimension to 128, batch size to 512, learning rate to 0.001, and negative sampling number to 1 for training and 199 for validation and testing. We used the Adam optimizer to update all parameters. For the comparison baselines, we adopted the hyper-parameter values reported in the official literature. In the case of SDSR models combined with our model, we did not modify the hyper-parameters of the SDSR models. Additionally, the threshold value to control the group flag $k$ is set to 0.7 and the size of the sampled users is set to the batch size. We set $\lambda_1$ = 0.01 and $\lambda_{2,3,4,5} = 1e^{-4}$. The learning rate for the first step with loss $\mathcal{L}_e$ is $1e^{-3}$ and the rate for the second step with loss $\mathcal{L}_r$ is $1e^{-5}$.

## C.2 Compared methods

**Single-domain sequential recommendation methods:**

**BERT4Rec** [38] designs a bidirectional self-attention network to model user behavior sequences. To prevent information leakage and optimize the training of the bidirectional model, a Cloze objective is used to predict the randomly masked items in the sequence by considering both their left and right context. The implementation of BERT4Rec in PyTorch can be found at the URL [4].

**GRU4Rec** [12] tackles the issue of modeling sparse sequential data while also adapting RNN models to the recommender system. To achieve this, the authors propose a new ranking loss function that is specifically designed for training these models. The implementation of GRU4Rec in PyTorch can be found at the URL [5].

**SASRec** [17] is a self-attention based sequential model that addresses the challenge of balancing model parsimony and complexity in recommendation systems. By using an attention mechanism, SASRec identifies relevant items in a user's action history and predicts the next item based on relatively few actions, while also capturing long-term semantics like an RNN. This enables SASRec to perform well in both extremely sparse and denser datasets. The implementation of SASRec in PyTorch can be found at the URL [6].

**Conventional Cross-domain recommendation methods:**

**STAR** [35] aims to train a single model to serve multiple domains by leveraging data from all domains simultaneously. The model captures the unique characteristics of each domain while also modeling the commonalities between different domains. It achieves this by using a network structure consisting of two factorized networks for each domain: one shared network that is common to all domains and one domain-specific network tailored to each domain. The weights of these two networks are combined to generate a unified network. The implementation of Pi-Net in Tensorflow can be found at the URL [7].

**MAMDR** [25] presents a novel model agnostic learning framework called MAMDR for multi-domain recommendation (MDR). It addresses the challenges of varying data distribution and conflicts between domains in MDR. MAMDR incorporates a Domain Negotiation strategy to alleviate conflicts and a Domain Regularization approach to improve parameter generalizability. It can be applied to any model structure for multi-domain recommendation. The work also includes a scalable MDR platform used in Taobao for serving thousands of domains without specialists. In the comparsion, we utilize their official implementation in Tensorflow, which can be found at the URL [8].

**SSCDR** [16] addresses the challenge of inferring preferences for cold-start users based on their preferences observed in other domains. SSCDR proposes a semi-supervised mapping approach that effectively learns the cross-domain relationship even with limited labeled data. It learns latent vectors for users and items in each domain and encodes their interactions as distances. The framework then trains a cross-domain mapping function using both labeled data from overlapping users and unlabeled data from all items. SSCDR also incorporates an effective inference technique that predicts latent vectors for cold-start users by aggregating their neighborhood information.

**Cross-domain sequential recommendation methods:**

**Pi-Net** [27] simultaneously generates recommendations for two domains and shares user behaviors at each timestamp to address the challenges of identifying different user behaviors under the same account and discriminating behaviors from one domain that could improve recommendations in another. By leveraging parallel information sharing, Pi-Net improves recommendation accuracy and efficiency for cross-domain scenarios in the Shared-account Cross-domain Sequential Recommendation task. The implementation of Pi-Net in Tensorflow can be found at the URL [9].

**DASL** [19] addresses the limitation of previous cross-domain sequential recommendation models by considering bidirectional latent relations of user preferences across source-target domain pairs, providing enhanced cross-domain CTR predictions for both domains simultaneously. The proposed approach features a dual learning mechanism and includes the dual Embedding and dual Attention components to extract user preferences in both domains and provide cross-domain recommendations through a dual-attention learning mechanism. The implementation of DASL in Tensorflow can be found at the URL [10].

---

[4] https://github.com/jaywonchung/BERT4Rec-VAE-Pytorch
[5] https://github.com/hungpthanh/GRU4REC-pytorch
[6] https://github.com/pmixer/SASRec.pytorch

[7] https://github.com/RManLuo/MAMDR/tree/master
[8] https://github.com/RManLuo/MAMDR/tree/master
[9] https://github.com/mamuyang/PINet
[10] https://github.com/lpworld/DASL

**C$^2$DSR** [2] enhances recommendation accuracy by addressing the bottleneck of the transferring module and jointly learning single- and cross-domain user preferences through leveraging intra- and inter-sequence item relationships. This approach overcomes the limitations of previous methods and captures precise user preferences. The implementation of C$^2$DSR in PyTorch can be found at the URL [11].

**Debiasing methods for recommendation:**

**DCRec** [47] is a new recommendation paradigm that claims to unify sequential pattern encoding with global collaborative relation modeling. It attempts to address the issues of label shortage and the inability of current contrastive learning methods to tackle popularity bias and disentangle user conformity and real interest. The implementation of DCRec in PyTorch can be found at the URL [12].

**CaseQ** [46] is proposed to alleviate the effects of popularity bias and temporal distribution shift in a single-domain sequential recommendation from training to testing. To achieve this, CaseQ employs a hierarchical branching structure combined with a learning objective based on backdoor adjustment, which enables the learning of context-specific representations. The implementation of CaseQ in PyTorch can be found at the URL [13].

**IPSCDR** [21] has developed a novel Inverse-Propensity-Score (IPS) estimator that is tailored for cross-domain scenarios. The approach also incorporates three types of restrictions for propensity score learning. By utilizing these methods, IPSCDR effectively alleviates domain biases, including selection bias and popularity bias, when transferring user information between domains. As there is no official code release available, we have reconstructed the code for IPSCDR by adapting a related IPS-based framework. The implementation of the IPS-based framework in PyTorch can be found at the URL [14].

### C.3 Hyperparameter Analysis

**The trade-off parameter** $\lambda_1$**.** To evaluate the impact of the trade-off parameter $\lambda_1$ in the loss function, we conduct a series of experiments with different values of $\lambda_1 = 0.001, 0.01, 0.1$ to search for the optimal value for our AMID model. The experiments are run five times and the results are reported by mean and variance. From the results in Table 7-8, it can be observed that the SDSR models with $\lambda = 0.01$ achieved the best performance on both the Cloth-Sport and Phone-Elec scenarios with different $\mathcal{K}_u$. Moreover, in the Cloth-Sport scenario, the models with $\lambda = 0.001$ perform better than those with $\lambda = 0.1$, while in the Phone-Elec scenario, the models with $\lambda = 0.1$ perform better than those with $\lambda = 0.001$.

**The threshold $k$ for the group.** We conduct ablation experiments by varying the threshold $k \in \{0.5, 0.6, 0.7, 0.8, 0.9\}$ to investigate the impact of the threshold $k$ on constructing interest groups in the multi-interest information module. We measure the average evaluation scores using NDCG@10 and HR@10 on the Cloth-Sport and Phone-Elec scenarios. Our results show that a larger threshold $k$ ($0.5 \rightarrow 0.7$) leads to better performance, as more related interest information can be transferred. However, when the

threshold $k$ is increased beyond 0.7, the model's performance drops due to noise interference and redundant information. Therefore, to achieve superior performance, we set the threshold $k$ to 0.7.

**The number of the sampled users.** To explore the impact of the number of sampled users on the multi-interest information module, we conduct ablation experiments varying the number of sampled users from 128 to 1024. We measure the average evaluation scores (NDCG@10 and HR@10) for both Cloth-Sport and Phone-Elec scenarios, and the results are shown in Figure 8. Our findings suggest that an increase in the number of sampled users initially improves the recommendation performance, but it eventually declines when the matching neighbors reach 1024. This observation indicates that having too few sampled users would provide insufficient transferred information, while too many sampled users could introduce interference noise and compromise the model's performance. In practice, we select the number of sampled users to be 512 as it results in the best performance for our model.

## D  POTENTIAL SOCIETAL IMPACTS

We propose an adaptive multi-interest debiasing framework to enhance the performance of most off-the-shelf SDSR methods. By utilizing our approach, e-commerce companies can recommend more relevant products to users and increase their revenue. Furthermore, our model AMID devises a debiasing framework that attends to minority users and explores their potential interests. As a result, our work promotes fairness in the recommender system and may help mitigate social inequality that may arise from algorithmic biases.

---

[11]https://github.com/cjx96/C2DSR
[12]https://github.com/HKUDS/DCRec
[13]https://github.com/chr26195/caseq
[14]https://github.com/samikhenissi/IPS_MF

**Table 7: Hyperparameter anaysis (%) of $\lambda_1$ on the bi-directional Cloth-Sport CDR scenario with different $\mathcal{K}_u$.**

| Methods | Cloth-domain recommendation | | | | Sport-domain recommendation | | | |
|---|---|---|---|---|---|---|---|---|
| | $\mathcal{K}_u$=25% | | $\mathcal{K}_u$=75% | | $\mathcal{K}_u$=25% | | $\mathcal{K}_u$=75% | |
| | NDCG@10 | HR@10 | NDCG@10 | HR@10 | NDCG@10 | HR@10 | NDCG@10 | HR@10 |
| $\lambda_1$ = **0.001** : | | | | | | | | |
| BERT4Rec [38] + AMID | 2.89±0.10 | 5.63±0.12 | 3.74±0.21 | 6.98±0.25 | 4.65±0.27 | 9.17±0.30 | 5.63±0.21 | 10.55±0.43 |
| GRU4Rec [12] + AMID | 3.01±0.15 | 5.82±0.19 | 3.79±0.13 | 7.20±0.32 | 4.77±0.11 | 9.32±0.31 | 5.81±0.22 | 10.91±0.25 |
| SASRec [17] + AMID | 3.08±0.29 | 6.05±0.35 | 4.11±0.09 | 7.50±0.27 | 4.88±0.19 | 9.36±0.25 | 5.87±0.22 | 10.94±0.20 |
| $\lambda_1$ = **0.01** : | | | | | | | | |
| BERT4Rec [38] + AMID | 2.99±0.07 | 5.70±0.13 | 3.79±0.15 | 7.09±0.32 | 4.73±0.29 | 9.30±0.41 | 5.70±0.12 | 10.63±0.36 |
| GRU4Rec [12] + AMID | 3.10±0.17 | 5.95±0.16 | 3.94±0.15 | 7.30±0.30 | 4.89±0.10 | 9.42±0.28 | 5.90±0.17 | 11.01±0.17 |
| SASRec [17] + AMID | **3.20±0.22** | **6.14±0.33** | **4.19±0.10** | **7.62±0.20** | **4.97±0.15** | **9.48±0.22** | **5.96±0.14** | **11.04±0.26** |
| $\lambda_1$ = **0.1** : | | | | | | | | |
| BERT4Rec [38] + AMID | 2.75±0.12 | 5.51±0.15 | 3.56±0.10 | 6.88±0.39 | 4.50±0.27 | 9.13±0.51 | 5.50±0.17 | 10.44±0.34 |
| GRU4Rec [12] + AMID | 2.91±0.19 | 5.76±0.13 | 3.79±0.14 | 7.11±0.29 | 4.68±0.09 | 9.24±0.34 | 5.73±0.15 | 10.79±0.31 |
| SASRec [17] + AMID | 3.02±0.21 | 5.97±0.28 | 3.99±0.15 | 7.40±0.17 | 4.75±0.11 | 9.25±0.30 | 5.77±0.19 | 10.83±0.24 |

**Table 8: Hyperparameter anaysis (%) of $\lambda_1$ on the bi-directional Phone-Elec CDR scenario with different $\mathcal{K}_u$.**

| Methods | Phone-domain recommendation | | | | Elec-domain recommendation | | | |
|---|---|---|---|---|---|---|---|---|
| | $\mathcal{K}_u$=25% | | $\mathcal{K}_u$=75% | | $\mathcal{K}_u$=25% | | $\mathcal{K}_u$=75% | |
| | NDCG@10 | HR@10 | NDCG@10 | HR@10 | NDCG@10 | HR@10 | NDCG@10 | HR@10 |
| $\lambda_1$ = **0.001** : | | | | | | | | |
| BERT4Rec [38] + AMID | 7.67±0.25 | 14.18±0.41 | 7.88±0.39 | 14.54±0.21 | 10.87±0.14 | 18.29±0.30 | 11.83±0.15 | 19.77±0.37 |
| GRU4Rec [12] + AMID | 8.05±0.13 | 15.20±0.45 | 8.08±0.10 | 15.02±0.23 | 11.44±0.10 | 19.21±0.14 | 12.25±0.15 | 20.58±0.25 |
| SASRec [17] + AMID | 7.94±0.11 | 14.69±0.09 | 8.05±0.32 | 14.68±0.23 | 11.40±0.29 | 19.03±0.24 | 12.21±0.15 | 20.48±0.19 |
| $\lambda_1$ = **0.01** : | | | | | | | | |
| BERT4Rec [38] + AMID | 7.95±0.23 | 14.49±0.36 | 8.15±0.33 | 14.85±0.28 | 11.26±0.07 | 18.58±0.27 | 12.16±0.09 | 20.08±0.31 |
| GRU4Rec [12] + AMID | **8.33±0.09** | **15.49±0.50** | **8.34±0.21** | **15.29±0.29** | **11.74±0.07** | **19.50±0.08** | **12.53±0.11** | **20.85±0.20** |
| SASRec [17] + AMID | 8.20±0.10 | 14.99±0.16 | 8.32±0.20 | 14.96±0.15 | 11.71±0.23 | 19.28±0.23 | 12.52±0.13 | 20.79±0.20 |
| $\lambda_1$ = **0.1** : | | | | | | | | |
| BERT4Rec [38] + AMID | 7.78±0.19 | 14.23±0.32 | 8.01±0.40 | 14.68±0.31 | 11.08±0.13 | 18.39±0.25 | 11.97±0.11 | 19.89±0.20 |
| GRU4Rec [12] + AMID | 8.14±0.15 | 15.28±0.39 | 8.15±0.19 | 15.12±0.25 | 11.55±0.11 | 19.35±0.10 | 12.41±0.16 | 20.66±0.22 |
| SASRec [17] + AMID | 8.05±0.21 | 14.83±0.13 | 8.15±0.19 | 14.78±0.20 | 11.52±0.15 | 19.08±0.19 | 12.33±0.10 | 20.62±0.12 |

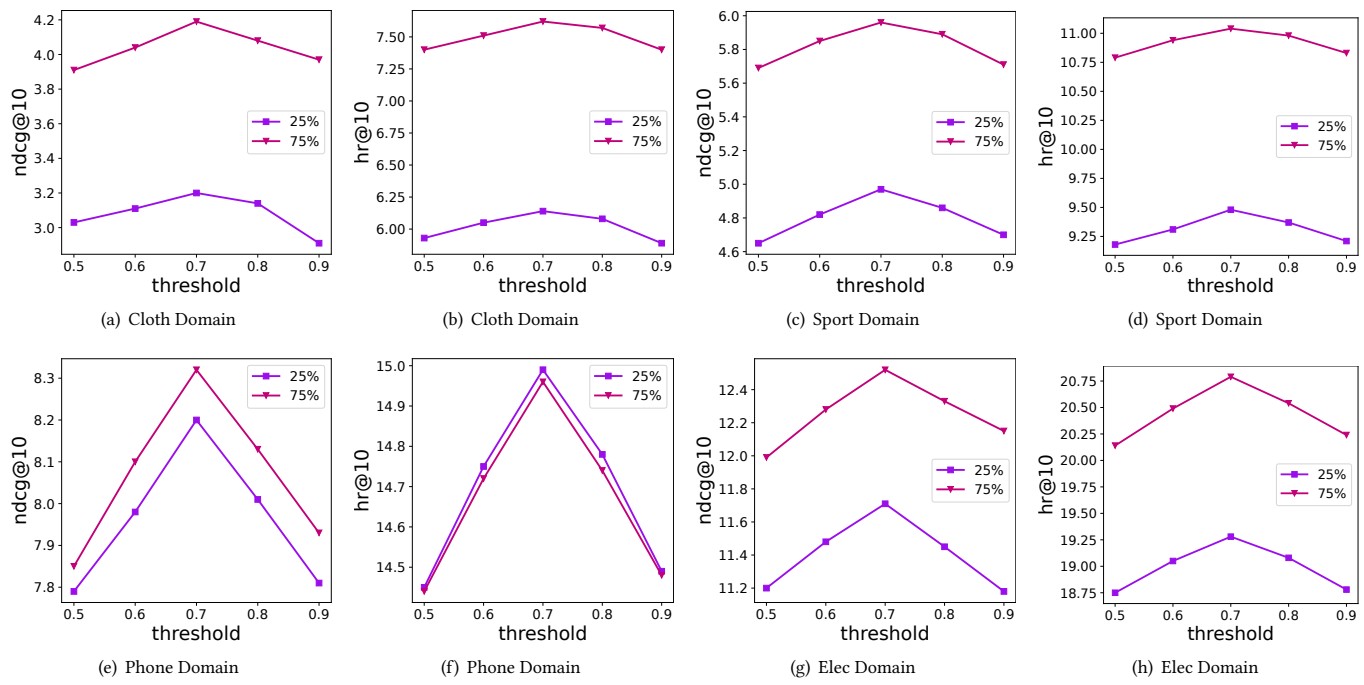

Figure 7: The results of SASRec+AMID with different threshold $k$ on the Cloth-Sport & Phone-Elec scenarios. 25% and 75% denotes different $\mathcal{K}_u$.

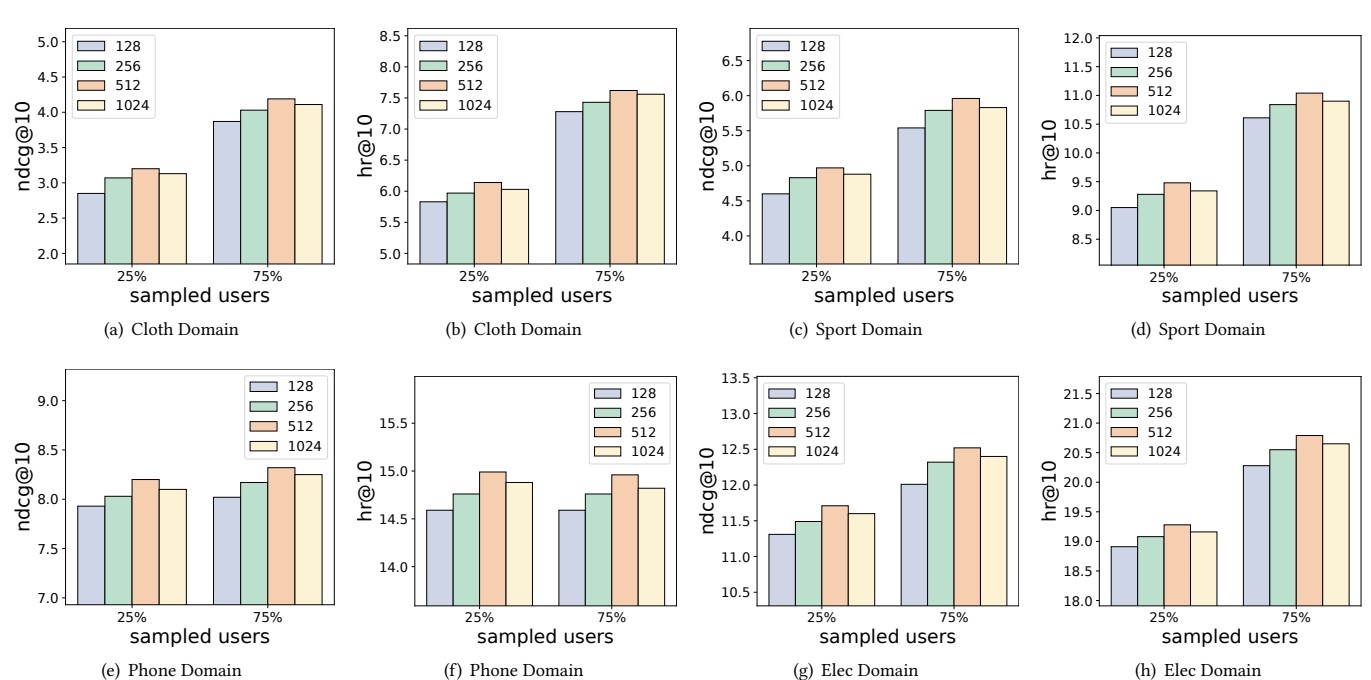

Figure 8: The results of SASRec+AMID with different number of the sampled users on the Cloth-Sport & Phone-Elec scenarios. 25% and 75% denotes different $\mathcal{K}_u$.

