# OpenReview forum: "Rethinking Cross-Domain Sequential Recommendation under Open-World Assumptions"
_ACM.org/TheWebConf/2024/Conference — TheWebConf24_

### Official Review · Reviewer_yobY · 2023-11-22

**Novelty:** 6
**Technical Quality:** 5

**Review:**

This manuscript addresses the challenges in cross-domain sequential recommendation methods under open-world assumptions. It identifies two primary challenges: how to construct a model in scenarios where most users are non-overlapping, and how to eliminate selection bias with data distribution shift. Extensive experiments with online A/B tests on a real-world CDSR financial platform demonstrate the effectiveness of AMID.

Advantages:

1. Adaptive Framework: AMID adapts to address the challenges of cross-domain sequential recommendation under open-world assumptions.
2. Doubly Robust Estimator (DRE): AMID utilizes a DRE to eliminate selection bias and popularity bias in the model.
3. Flexibility: AMID can be integrated with most sequential recommendation backbone models.
4. Superior Performance.

Shortcomings:

1. Assuming a high level of prior knowledge without providing sufficient background information.
2. Lack of some model details.
3. Experiments of the non-overlapping user scenario.

**Questions:**

[Regarding Motivations]

1. I find the author's explanation of the motivations somewhat perplexing. Can the author clarify the distinctions of CDR with many overlapping users, those with a few overlapping users, and those without overlapping users?
2. Could you provide a precise definition of open-world cross-domain recommendation? Does it refer to the combination of domain distribution shift and partial overlapping scenarios?

[Regarding the Objective Function]

From my understanding, the objective function in Eq. 1 seems incomplete. When predicting interactions for users in the source domain $X$, it only considers user features without utilizing user interaction sequences in the target domain $Y$.
How does CDR obtain comprehensive user representations in the target domain without using the interaction sequences of users?

[Regarding Doubly Robust Estimator]

1. The specific details of how the propensities are computed are not explicitly mentioned in the provided context.
2. The manuscript lacks explicit details about the model frameworks for the imputation model and propensity model.

[Regarding Joint Learning]

1. The label y_{u,v} does not appear in the optimization objective.
2. Is the same user represented by the same user representation across different domains? Or does it need adjustment based on each domain?

[Regarding Experiments]

The authors mention the feasibility of the model in a non-overlapping scenario, which need to be validated by the experiments.

[Other]

It appears that the authors assume a high level of prior knowledge without providing sufficient background information. Readers unfamiliar with the specific domain may find it challenging to grasp the context and concepts.

**Reviewer Confidence:**

3: The reviewer is confident but not certain that the evaluation is correct

**Scope:**

4: The work is relevant to the Web and to the track, and is of broad interest to the community

---

### Official Review · Reviewer_MFqN · 2023-11-22

**Novelty:** 5
**Technical Quality:** 5

**Review:**

This study targets CDSR under open-world assumption with fewer overlapping users and shifting distributions. Specifially, the authors focus on the gap in modeling non-overlapping users and addressing domain-related selection bias, introducing a multi-interest debiasing framework compatible with current off-the-shelf methods. Empirical investigations include the collection of real-world financial CDSR datasets, offline method comparisons and online A/B tests.

**pros**

1. The problem being tackled is indeed more aligned to real-world scenarios compared with many existing CDR studies
2. Technically sound methodology with formal guarantee on bias and variance
3. The proposed MYbank-CDR, if become available to the public, is expected to enable more research

**cons**

1. User grouping is based solely on sequence similarity. This pure data-driven process would potentially lead to instability and bias.  I wonder if the authors have considered or tried to incorporate any kind of domain knowledge.
2. lack of an explicit section discussing related literature - I would have to say it is difficult to evaluate the novelty (in methodology) in this case.

**Questions:**

1. I wonder if the authors have considered or tried to incorporate any kind of domain knowledge for user group construction.
2. I wonder will the authors supplement more discussions about related studies. I would refrain to grade a high grade on "novelty" in its current form, and will open to adjustment upon the authors' reply.

**Ethics Review Description:**

no ethical issues were identified

**Reviewer Confidence:**

4: The reviewer is certain that the evaluation is correct and very familiar with the relevant literature

**Scope:**

4: The work is relevant to the Web and to the track, and is of broad interest to the community

---

### Official Review · Reviewer_SonC · 2023-11-23

**Novelty:** 4
**Technical Quality:** 6

**Review:**

### Pros
1. The selected topic: the debias problem under the CDSR settings is interesting.
2. The evaluation process of the paper is rigorous and comprehensive.

### Cons
1. The reviewer suggests revising the paper's citations. Multi-interest recommendation has been a topic studied since 2019, yet the author cites none of the previous work. This is the same for the usage of causal graphs in Section 3.2.
2. Some of the methodology sections are not self-contained. Parameters are used without explicit introduction. Check minors and questions for more details.
3. The reviewer also suggests the author anonymize the online platform or private dataset, as it may potentially violate the double-blind principle. Otherwise, the industrial track seems to be a more suitable option.
4. The paper introduces the multi-interest methods into the cross-domain recommendation to solve the non-overlapping issue and adopts double robustness for debiasing. However, this appears to be a stack of existing technologies without any novel contribution to the community.

### Minors:
1. Error in line 304 without the rating set
2. Y and P^X in Section 3.1. are never properly introduced
3. sigma in Eq. 7 is never properly introduced

**Questions:**

1. What's the meaning of Eq 1 and 2? Why is equalizing the rating r and item v in Eq 1 while Eq 2?
2. The reviewer would like to learn about the difference between the current paper and previous works[1,2]
3. The reviewer would also like the author to include one part in related work regarding multi-interest recommendations.

[1] Joint Internal Multi-Interest Exploration and External Domain Alignment for Cross Domain Sequential Recommendation, WWW 2023

[2] M5: Multi-Modal Multi-Interest Multi-Scenario Matching for Over-the-Top Recommendation, KDD 2023

**Reviewer Confidence:**

3: The reviewer is confident but not certain that the evaluation is correct

**Scope:**

3: The work is somewhat relevant to the Web and to the track, and is of narrow interest to a sub-community

---

### Official Review · Reviewer_zwuB · 2023-11-24

**Novelty:** 6
**Technical Quality:** 6

**Review:**

The paper proposes a novel method for cross-domain recommendation. First, the authors summarize two main challenges of cross-domain recommendation: few overlapping users and model's selection bias. To overcome those challenges, two mechanisms are employed. Multi-interest information module groups users in all domains and leverage information from users in the same group. A doubly robust estimator is then used to imitate the bias problem. A lot of experiments are conducted to demonstrate the effectiveness of the method, including both public dataset, industrial dataset and online test.


### Strengths
1. The cross-domain and bias problem are extremely important in recommendation field, both for research and industry. The paper demonstrates thoughtful considerations and analysis of the issue, presenting an innovative approach to address the problems.
2. The used doubly robust estimator has good theoretical properties in debiasing. Theoretical analysis is provided to demonstrate those qualities.
3. Plenty of experiments including evaluation on industrial dataset and online test are conducted, demonstraing effectiveness of the proposed MIM and AMID methods.


### Weaknesses
1. Some symbols used in equations lack of definition and explaination, making it difficult to understand. For example, $o,\delta, \hat{p}$ in equation 9.
2. The author claimed the challenges in cross domain under open-world assumptions. But the experimental settings keep as a closed-world assumption, where there are many overlapping users.
3. The time and space complexity of the method maybe a concern. The complexity in MIM would be $O(N^2)$ (N is the number of users). For multiple domains, the cost may be not acceptable.

**Questions:**

1. Could you give some complexity analysis of the MIM? I concern it would be the bottleneck to extend to multiple domains.
2. It is strange that the performance of BERT4Rec reported in table1 is surpassed by GRU4Rec and SASRec, which is inconsistent with the results in previous studies. Can you give an explanation about the results?

**Reviewer Confidence:**

2: The reviewer is willing to defend the evaluation, but it is likely that the reviewer did not understand parts of the paper

**Scope:**

4: The work is relevant to the Web and to the track, and is of broad interest to the community

---

### Official Review · Reviewer_ry5K · 2023-11-26

**Novelty:** 6
**Technical Quality:** 6

**Review:**

This work addresses the challenges in Cross-Domain Sequential Recommendation (CDSR) by proposing an Adaptive Multi-Interest Debiasing framework (AMID) under open-world assumptions. Unlike existing CDSR methods that assume fully overlapping users across domains and consistent data distribution, AMID is designed to adapt e to open-world settings, enhancing performance on real-world platforms with data distribution shifts. The framework comprises a multi-interest information module (MIM) and a doubly robust estimator (DRE). This work is well-motivated, and the proposed solution sounds reasonable.

Strong points:
S1. This work addresses solving cross-domain sequence recommendations under open-world settings, which is more challenging than the trandational methods.
S2. This work is well-motivated, and the proposed solution sounds reasonable.
S3 This work has conducted extensive experiments, and the results are impressive.

Weak points:
W1. I have some questions about the trend of changes in (a) and (b) in Figure 2. It is normal for cross-domain methods to experience performance fluctuations due to changes in overlapping user ratios, but why do single-domain methods also experience significant performance fluctuations? Generally, the single-domain recommendation method uses data from only one domain.
W2. The third paragraph of the introduction summarizes C2DSR as a non-overlapping method, but in related work, C2DSR is also classified as a method that relies on overlapping entities. Is this contradictory?
W3. The author's definition of the open world hypothesis is relatively vague, and it is suggested to rephrase the concepts of open and closed worlds in plain language。

**Questions:**

Weak points 1-3.

**Reviewer Confidence:**

3: The reviewer is confident but not certain that the evaluation is correct

**Scope:**

3: The work is somewhat relevant to the Web and to the track, and is of narrow interest to a sub-community

---

### Decision · Program_Chairs · 2024-01-22

**Decision:**

Accept

**Comment:**

The paper presents a novel approach to Cross-Domain Sequential Recommendation (CDSR) by introducing an Adaptive Multi-Interest Debiasing framework (AMID) under open-world assumptions. This innovative approach is designed to adapt to real-world platforms where data distribution shifts are common, making it a significant advancement in the field of recommendation systems. The framework's integration of a multi-interest information module (MIM) and a doubly robust estimator (DRE) is a commendable effort in addressing the challenges in CDSR. The key strengths of this work lie in its relevance to real-world scenarios, its well-motivated approach, and the extensive experimental validation, which includes both public and industrial datasets as well as online tests. The proposed solution demonstrates impressive results, indicating its effectiveness and practical applicability. Theoretical analyses provided in the paper further reinforce the robustness of the methodology. There are areas where the paper could be improved for enhanced clarity and comprehensiveness. The definitions and explanations of some concepts, particularly the open-world hypothesis, need to be more explicit and accessible. The paper would benefit from a more detailed discussion of related works, especially in the context of multi-interest recommendations and the use of causal graphs. Additionally, certain technical aspects, such as symbol definitions in equations and the methodology's time and space complexity, require clearer exposition.